# Transcriptomic dynamics reveals sequential acquisition of complement resistance during prolonged starvation of *Trypanosoma cruzi* epimastigote

**Leticia Pérez-Díaz¹, Pablo Smircich¹,², Fabricio Hernandez¹, Martin Ciganda¹, Ma Ana Duhagon¹,³, Beatriz Garat¹/+**

¹Facultad de Ciencias, Sección Genómica Funcional, Montevideo, Uruguay
²Instituto de Investigaciones Biológicas Clemente Estable, Departamento de Genómica, Montevideo, Uruguay
³Facultad de Medicina, Departamento de Genética, Montevideo, Uruguay

**BACKGROUND** The life cycle of the parasitic protozoan *Trypanosoma cruzi*, the etiological agent of Chagas disease (CD), includes two well-recognised insect-dwelling stages: the replicative non-infective epimastigotes and the non-replicative infective metacyclic trypomastigotes. Nonetheless, the existence of multiple intermediate forms has been reported. Since nutrient restriction is considered one of the main factors driving metacyclogenesis and is very frequent due to the long-term starvation periods that the insect vectors commonly undergo, we have studied the transcriptomic effects of nutrient restriction on long-lasting epimastigote cultures. We previously reported that in these conditions, we observed a long stationary phase characterised by an RNA content per cell three times smaller than the epimastigote's and a distinctive transcriptomic profile. Remarkably, our study identified gene expression changes that distincty characterise transitional parasite forms enriched by nutrient restriction.

**OBJECTIVES** In this work we focused on pathogenic genes to further characterise the transcriptomic dynamics accompanying the nutrient restriction within the insect-dwelling parasite stage.

**METHODS** The alterations of morphology, growth rate and complement resistance of parasite population on long-lasting epimastigote cultures as well as the transcriptomic dynamics was studied.

**FINDINGS** We found a gene expression early rise of surface proteins (such as trans-sialidase and GP63) and even a rise of TcTASV and δ-amastin, which is not accompanied by increased expression of metacyclic transcript markers. In addition, we found increased expression of genes coding for proteins involved in two other processes activated during the differentiation of epimastigotes to the infective form of the parasite: autophagy (Atg4, Atg7, Atg8.2) and complement resistance (TcCRP and T-DAF).

**MAIN CONCLUSIONS** Altogether, these results, plus our previous identification of transcriptomic markers for transitional parasites, further support earlier proposals of a specific parasite stage that morphologically resembles epimastigotes but exhibits distinctive biological characteristics, including key features related to infectivity.

Key words: *Trypanosoma cruzi* - life cycle development - transcriptomics

*Trypanosoma cruzi* (Kinetoplastidae, Trypanosomatidae) is a parasitic protozoan that causes Chagas disease (CD),[1] a major socio-economic problem in Latin America, where it is considered endemic in 21 countries, with circa 8 million people infected and about 100 million at risk of infection.[2] The parasite is transmitted by blood-sucking triatomines widely distributed in Latin America. Since the parasite can also be transmitted by contaminated food, congenitally from mother to child and through contaminated blood or organ donations, CD has spread to non-endemic areas such as North America, Europe, and the western Pacific, due to migratory flows.[3]

As described by Chagas, the parasite has different stages along its complex life cycle.[1] At least four stages alternating between triatomine vectors (*Triatoma in-festans*, Hemiptera, Reduviidae), and mammalian hosts are currently accepted. The non-infective epimastigote form, which actively replicates in the vector's midgut, differentiates into non-replicative metacyclic trypomastigotes in the hindgut. These forms are deposited with faeces and are responsible for the infection of mammals. In the mammalian hosts, the metacyclic trypomastigotes infect cells differentiating into intracellular replicative amastigotes, which finally differentiate into the blood non-replicative trypomastigotes that, after cell lysis, can invade other cells and tissues, producing the clinical manifestations of CD.

The parasite transition from epimastigotes to the infective metacyclic trypomastigote is a critical biological step in establishing the infection. This process, known

Financial support: This work was supported by CSIC I+D Grupo 108725, UdelaR and PEDECIBA, Uruguay.

+ Corresponding author: bgarat@fcien.edu.uy | ⬥ https://orcid.org/0000-0003-2679-8244

**Handling editor:** Adeilton Alves Brandão | ⬥ https://orcid.org/0000-0001-5877-607X

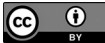

as metacyclogenesis, takes place along the insect rectum, where parasites are pulled through increasingly nutrient-restricted environments. The reduced nutrient availability is the primary stimulus to induce processes such as autophagy, whose participation in the parasite differentiation process has been established.[4,5] Several morphological and structural features distinguish insect epimastigotes from metacyclic trypomastigotes. These include modifications of the relative nucleus-kinetoplast location, elongation of the cytoplasm, nucleolar disaggregation, dispersed content of heterochromatin, increase in the flagellum pocket size, and several concomitant metabolic and physiological changes.[6,7,8] Metacyclic trypomastigotes also show an increased expression of proteins associated with virulence [*e.g.*, GP63, mucins, and mucin-associated surface proteins (MASP)], and the specific metacyclic protein marker metacyclins (Met-II and Met-III).[9,10,11] In contrast to metacyclic trypomastigotes, epimastigotes are highly susceptible to complement-mediated lysis[12] though the ability to resist the complement differs among different *T. cruzi* strains.[13] It has been early recognised that stage specific gene expression precedes morphological changes during metacyclogenesis.[14]

Despite constitutive transcription of protein-coding genes in kinetoplastids, significant transcriptomic changes between epimastigotes and metacyclic trypomastigote have been reported by us and others.[15,16,17,18,19,20] Remarkably, using traslatomics, we established translation efficiency as a crucial developmental regulatory step along this transition.[15] Additionally, significant proteomic differences between these two developmental stages were reported.[21,22] Indeed, the analysis of *in vitro* cultured exponential and initial stationary phase epimastigotes has revealed either expression changes of genes related to cell cycle, pathogenesis, and metabolic processes[23] as well as proteins related to the replication status[24] and metabolic switch from glucose to amino acid consumption in stationary phase epimastigotes.[25] Consistently, a recent metabolome study of nutritional and oxidative stress supports the rapid *T. cruzi* adaptation to environmental changes.[26] While these molecular changes may reflect the beginning of the metacyclogenesis process, the existence of an intermediate distinctive preadaptive stage with the ability to differentiate either into the metacyclic form or to return to the replicative epimastigote stage depending on the availability of nutrients has been proposed.[27] It is well recognised that the enormous amount of blood ingested and the long period of starvation affect the intestinal environments in the vector, yielding different proportions of developmental stages, including not only epimastigotes and metacyclic trypomastigotes but also many other intermediate parasite forms.[28] Nutritional stress during the *in vitro* metacyclogenesis process also leads to the appearance of different intermediate parasite forms.[7,29,30,31,32] These forms have been globally named "transitional epimastigotes".[32] The plasticity and complexity of *T. cruzi* forms along the life cycle spread beyond the epimastigote to metacyclic trypomastigote transition. Transient *T. cruzi* epimastigote-like forms

as intermediates in the differentiation of amastigotes to trypomastigotes inside the mammalian host cells[33] and their distinct energy and carbon source requirements compared to the other intracellular stages[34] have been characterised. In addition, the differentiation from the trypomastigote forms (cell-derived or metacyclic) to an epimastigote-like form named "recently differentiated epimastigotes", exhibiting complement resistance and infection ability has been recently described using cell biology and proteomic approaches.[35]

To understand the molecular changes caused by nutritional restrictions in the insect host, we have recently reported the transcriptomic changes during axenic growth of epimastigotes for more than 30 days without nutrient supplementation (prolonged starvation).[36] In these conditions, we observed an extended stationary phase characterised by an RNA content per cell three times smaller than that of epimastigotes. This parasite population exhibited a distinctive transcriptomic profile. Ontology-enriched terms for cellular components such as contractile vacuole, reservosomes, and the mitochondria were revealed, suggesting a protagonistic role possibly related to their functions in osmoregulation and metabolic homeostasis and cell volume regulation for the adaptation to the nutrient restriction. In this parasite population, we also found a distinctive expression of genes related to DNA, granting the quiescent status in starving conditions. Remarkably, our study identified differentially expressed genes (DEGs) that constitute markers of this transitional parasite population enriched by nutrient restriction, supporting the existence of a distinctive stage between the recognised insect-dwelling forms.[27,29,32]

To further characterise the complex molecular dynamics accompanying the nutrient restriction within the insect branch of *T. cruzi* life cycle, we here focus on the transcriptomic analysis of pathogenic gene dynamics within the long stationary phase of axenic epimastigote culture in prolonged starvation. Firstly, we analysed the morphological characteristics of this parasite population. An increasing proportion of intermediate parasite forms with the nucleus-kinetoplast location characteristic of epimastigote and different growth resume ability was found. In addition, this parasite population exhibits an early increase of genes involved in surface protein genes, which is not accompanied by increased expression of some metacyclic transcript markers such as metacyclin II and III. Considering the involvement of surface proteins in infectivity, we focused on two other processes activated during the differentiation of epimastigotes to the infective form of the parasite: autophagy and complement resistance. We found increased expression of genes related to both these processes and the complement resistance ability was also experimentally verified. These results complement the distinctive transcriptomic profile we previously reported for transitional parasites obtained along the axenic growth of *T. cruzi* epimastigotes for over 30 days without nutrient supplementation[36] and further support previous proposals[32] regarding the existence of a specific parasite stage morphologically resembling epimastigotes but exhibiting distinct biological characteristics.

## MATERIALS AND METHODS

*Parasite culture and morphology analyses* - *T. cruzi* Dm28c strain (TcI DTU) epimastigotes were cultured as previously indicated.[36] Three biological replicates were assessed to determine the growth curve dynamics. Parasites were directly counted by light microscopy using a Neubauer chamber, and triplicates for each independent experiment were analysed. Occasionally, parasite concentration was also verified by flow cytometry using a BD Accuri C6. Fluorescent microscopy (Leica TCS-SP5 and ZOE Fluorescent Cell Imager) was used to analyse images of 4',6-diamidino-2-phenylindole (DAPI) of paraformaldehyde (PFA) fixed parasites. Several images were acquired, and at least 100 cells were counted in each triplicate for each independent biological replica.

*Transcriptomic analysis* - Expression data was obtained from.[36] Genes were considered differentially expressed if they exhibited a log2 fold change (FC) of |1| and a false discovery rate (FDR) less than 0.05. Over-representation of GO terms among the differentially expressed genes was determined using the tools available at TritrypDB (http://tritrypdb.org/). A Bonferroni adjusted p-value of less than 0.5 was used as the significance cutoff. Unless stated otherwise, statistical analysis and plotting were conducted in R.

*Sensibility to human serum complement* - Parasites in the exponential growth phase ($2 \times 10^7$ parasites/mL) were diluted in culture media (1/10) containing 10% human serum that had or had not been heat treated (60ºC for 15 min). In all cases, cell viability was analysed after 24 h of incubation with the serum. The cellular viability and vitality were assessed through Propidium Iodide (PI) and Calcein-AM (CA) labelling, respectively followed by flow cytometry using a 670 nm band-pass filter (FL3) (Accuri C6, BD Bioscience). For labelling, parasites were incubated with $1 \times 10^{-6}$ M CA for 1 h and 10 mg/mL PI (Thermo Fisher Scientific) for 15 min at RT in the darkness. At least three independent samples were assayed for each condition, and 10,000 events were acquired per experiment.

## RESULTS AND DISCUSSION

*Morphology and growth kinetics during prolonged starvation of T. cruzi epimastigote* - Along the growth curve of *in vitro* cultured *T. cruzi* epimastigotes for more than 30 days,[36] the successive lag phase followed by the exponential, the stationary phase and the final death phase were observed [Supplementary data (Fig. 1A)]. The exponentially growing parasite population mostly exhibits the characteristic spindle-shaped cells and normal flagellar motility of epimastigotes (Fig. 1 top panel), although a small percentage (< 5%) of cells present a sickled-shaped morphology resembling metacyclic trypomastigotes (Fig. 1 bottom panel). Morphological differences between epimastigotes and metacyclic trypomastigotes also include the position of the kinetoplast, being anterior in the epimastigote and posterior in the metacyclic trypomastigote (compare the 2nd and 3rd columns of top and bottom panel in Fig. 1). During the stationary phase, the parasites become longer and slender,

maintaining the position of the kinetoplast and flagellum base relative to the nucleus as in the epimastigote stage [Fig. 1 middle panels and Supplementary data (Fig. 1B)]. This elongation of the body and flagellum in stationary phase is well-known.[27,29,32] Flagellar elongation may provide an extended nutrient uptake surface in unfavourable nutrient conditions[29] and may constitute an early step driving the flagellar attachment required for metacyclic development.[37] Consequently, repositioning of the kinetoplast and loss of endocytic ability have only been observed in the later stages of the metacyclogenesis process.[6,7,38] However, an increasing distance between the nucleus and the kinetoplast was observed. In these conditions, we have found that the percentage of metacyclic trypomastigote within this long stationary phase displays a composed profile including a gradual increase, not surpassing 10% for a long period (3.8 ± 1.7%, 5.4± 0.4% and 7.9 ± 0.6% for the exponential phase -day 7-, early stationary phase -day 14-, and intermediate stationary phase-day 21 respectively) and then a sharp increase at the end (32.1 ± 5.4% for the final of the stationary phase -day 28-).[36] In accordance with the distinctive capacity to resume growth of epimastigotes, metacyclic trypomastigotes, and preadaptive forms,[27] all the analysed parasite populations were able to resume growth when the medium was replaced with fresh nutrients [Supplementary data (Fig. 1C)]. As anticipated, according to the expected doubling time of approximately one day (21.2 ± 0.7 h), parasites in the exponential growth phase are the fastest (0.79 parasites/mL.day). Afterwards, the older the parasites, the slower the growth rate.

In summary, in the assayed conditions of prolonged starvation of axenic culture of *T. cruzi* we observed the reported co-existence of multiple intermediate parasite forms.

*Dynamic analysis of transcriptomic changes during prolonged starvation of T. cruzi epimastigote* - We have previously reported that parasites in the long stationary phase provoked by prolonged starvation of epimastigote culture show a distinctive transcriptomic profile with defined DEGs markers.[36] Nonetheless, a gradual variation of some DEGs was also noted. To study the transcriptome dynamics of the parasite population along the prolonged starvation of the axenic culture of *T. cruzi* epimastigotes, we here present the analysis of the previously reported transcriptomic data.[36]

The comparative analysis of DEGs between the selected time points: exponential phase (day 7, D7), early stationary phase (day 14, D14), intermediate stationary phase (day 21, D21), and final stationary phase (day 28, D28) is shown in Fig. 2. In order to distinguish temporal processes triggered by prolonged starvation within the long stationary phase of *T. cruzi* epimastigote culture, DEGs were classified as nutrient restriction early response transcripts (ERT) if significant different expression between the data derived from epimastigotes in the exponential (D7) and in the early stationary phase (D14) was observed (subclassified in Fig. 2 as following: D14 vs D7, D14 vs D7 plus D21 vs D7, D14 vs D7 and D21 vs D7 and D28 vs D7), likewise, DEGs were classified as late response transcripts (LRT) if the dif-

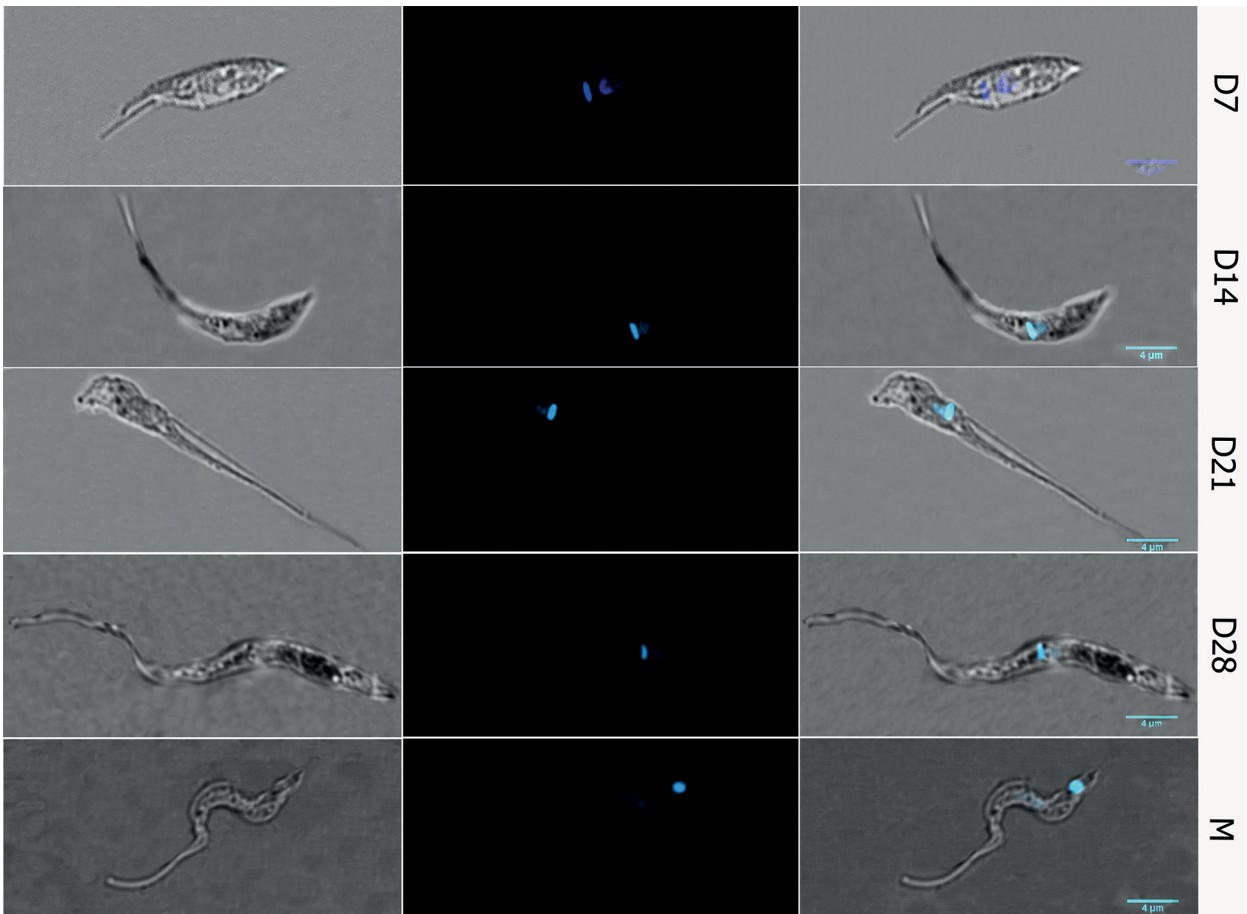

Fig. 1: morphology of *Trypanosoma cruzi* epimastigotes in prolonged starvation cell culture. Representative images of parasites observed by Confocal microscopy Leica TCS-SP5 100X.Top and bottom panels: epimastigote (D7) and metacyclic trypomastigote-like parasite (M) forms observed in exponentially growing epimastigote cultures. Middle panels: intermediate forms observed in starved cell cultures at days: 14, 21 and 28 (D14, D21 and D28 respectively). Right panel: phase contrast, central panel: DAPI stained parasites, left panel: overlay of both images.

ferent expression was restricted to the changes between intermediate and late stationary phase (D28 vs D21) [Supplementary data (Fig. 2, Table I)]. GO analysis for ERT and LRT DEGs is shown in Fig. 3. Upregulated ERT show an enrichment of genes coding for proteins involved in cell adhesion and glycosyl bond hydrolase activity, while the downregulated ERT are enriched in genes coding for proteins involved in carbohydrate and small molecule metabolic process, protein folding, chromosome organisation, lyase activity, unfolded protein binding, oxidoreductase and isomerase activity and DNA binding. For LRT, we found an upregulation of genes for proteins with hydrolase activity of glycosyl bonds and a downregulation of genes coding for proteins involved in cell population proliferation, lipid metabolic processes, and ATPase activity. It is worth noting that the profiles of previously reported markers of the metacyclic trypomastigotes, genes coding for Metacyclin II (TcCLB.506529.600) and Metacyclin III (TcCLB.509251.6)[9,39] remain constant all along the prolonged starvation of the *T. cruzi* epimastigote culture not accompanying the slow increase of metacyclic forms described above [Supplementary data (Fig. 2)].

The GO terms enriched in the LRT are consistent with the quiescent status of the epimastigotes in the stationary phase and the decreased doubling time observed for the D28 parasite population [Supplementary data (Fig. 1)]. On the other hand, the GO terms enriched in the ERT can be interpreted as a rapid parasite adaptation triggered by incipient nutrient shortage that promotes the differentiation of parasites to infective stages. These findings prompt us to focus on the expression dynamics of protein-coding genes involved in adhesion, infectivity, and complement resistance.

*Surface protein gene expression during prolonged starvation of T. cruzi epimastigote* - Surface proteins, arranged as large gene families exhibiting diversification and copy number diversity,[40] play different roles in the parasite's life cycle progression, in the host-cell interplay, immune system evasion and persistence of the parasite.[41]

Among them, the surface proteins mucins and MASPs constitute a highly expressed gene family in *T. cruzi*[42,43] involved in the parasite resistance against host immune system and attachment to the host cells.[44,45] Considering sequence similarities, the mucin family can be classified in TcMUC and TcSMUG.[44,45]

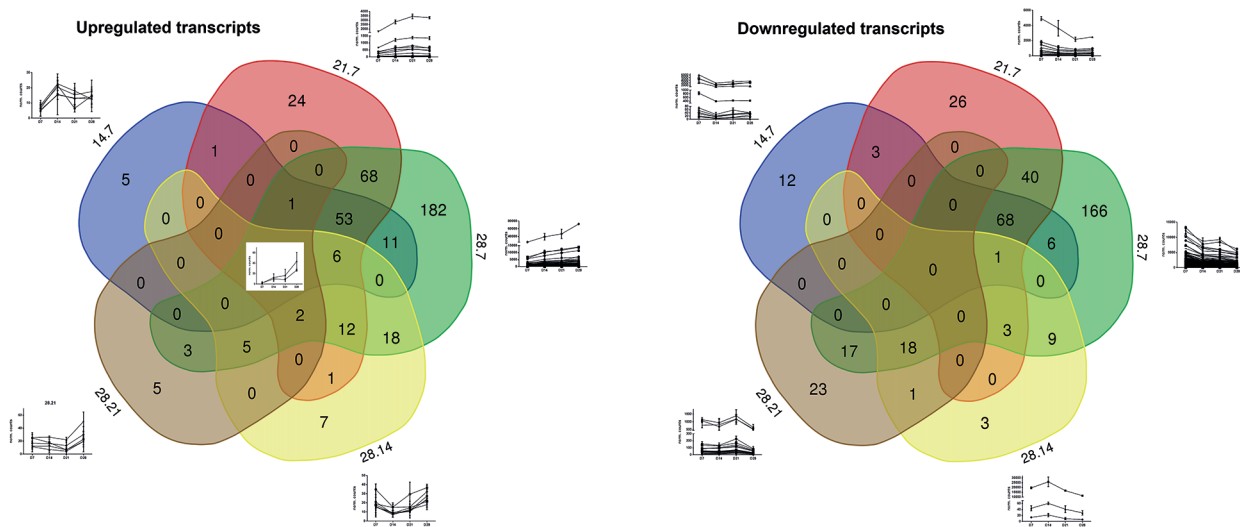

Fig. 2: Venn diagrams showing the number of common differentially expressed genes (DEGs) between the different time points during prolonged starvation of *Trypanosoma cruzi* epimastigote culture. The time points selected during the prolonged starvation of *T. cruzi* epimastigote culture correspond to the exponential phase: day 7 (7); early stationary phase: day 14 (14); intermediate stationary phase: day 21 (21); and the final of the stationary phase: day 28 (28). The expression profile of DEGs belonging to the indicated compartments is shown in each graph as the mean of the normalised read count with its standard error at each time point from:[36] 14.7 (Se vs E); 21.7 (Si vs E); 28.7 (Sf vs E); 28.14 (Sf vs Se); 28.21 (Sf vs Si).

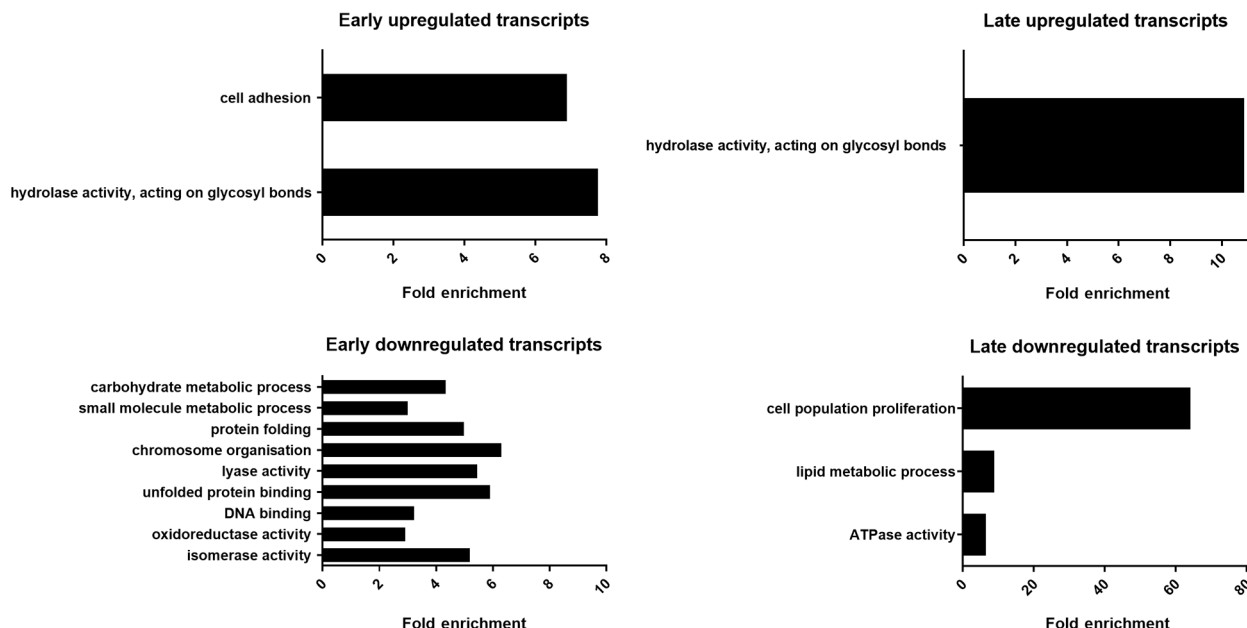

Fig. 3: gene ontology (GO) analysis of early and late modulated transcripts (ERT and LRT respectively) identified during prolonged starvation of *Trypanosoma cruzi* epimastigote cultures. GO enrichment analysis (using GO Slim terms) was performed on TriTrypDB for early and late response DEGs up (upper panels) and down (bottom panels) regulated. The most significantly ($p < 0.05$) enriched GO terms in biological process and molecular function branches are presented.

While the TcMUC subfamily expression, as well as the MASPs', are restricted to the parasite stages in the mammalian host,[46,47] the TcSMUG subfamily, is mainly expressed in the insect-dwelling stages.[48,49] The TcSMUG subfamily, in turn, is divided according to the size into two groups: small (S), composed of the 35-50 kDa mucins found in epimastigotes and metacy-clic trypomastigotes, and large (L) which are not sialic acid acceptors and are only present in the surface of the epimastigote stage.[49] All the annotated TcSMUG significantly diminish their expression, being TcSMUGL mostly ERT (2 out of 3) and maintain or exacerbate the profile along the long-lasting culture of *T. cruzi* [Fig. 4, Supplementary data (Table II)].

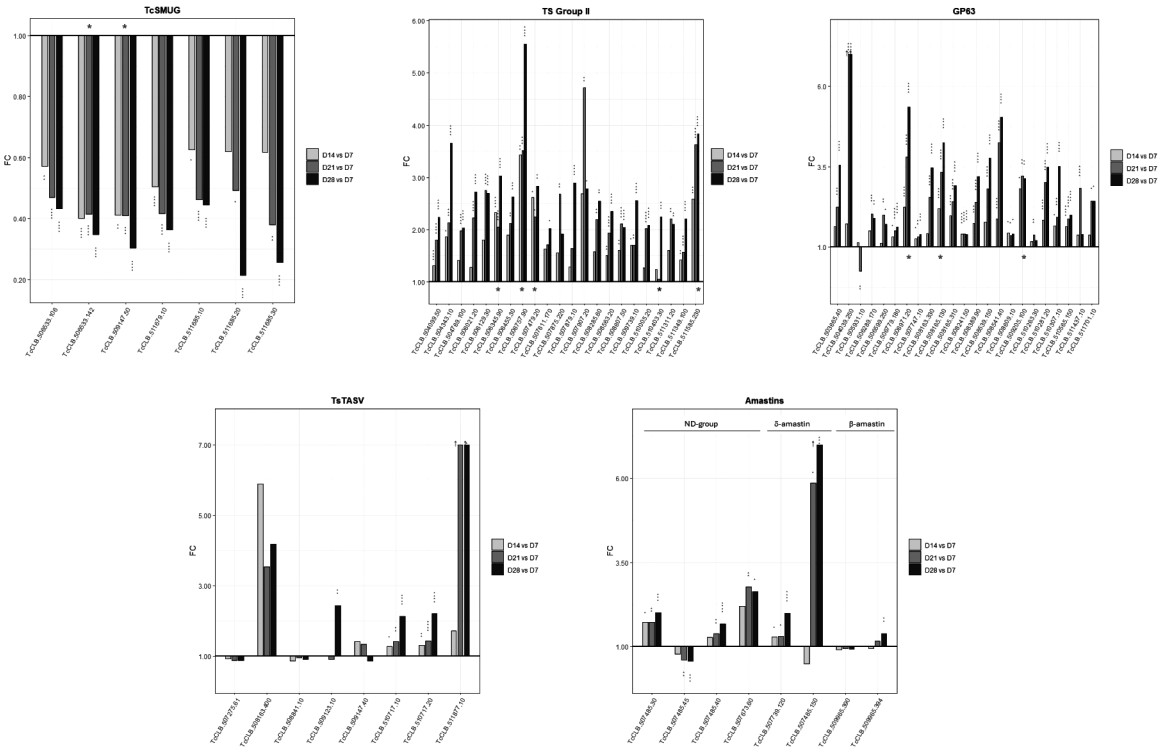

Fig. 4: differential expression of transcripts coding for surface proteins identified during prolonged starvation of *Trypanosoma cruzi* epimastigote culture. The expression profile of genes of the indicated families of surface protein is shown. During the prolonged starvation of *T. cruzi* epimastigote culture time points at day 7, corresponding to the exponential phase, day 14, early stationary phase, day 21, intermediate stationary phase and day 28, the final of the stationary phase (D7, D14, D21 and D28 respectively) were selected for analysis. Light grey bars represent the expression at D14 relative to D7; grey bars the expression at D21 relative to D7 and black bars the expression at D28 relative to D7. Vertical asterisks over each bar indicate adjusted significance: * p < 0.05, ** p < 0.01, *** p < 0.001. Red and blue asterisks account for ERT and LRT, respectively. The arrow (↑) indicates the bar was truncated at FC = 7.

A different profile was found for the highly expressed trans-sialidase (TS) superfamily, one of the most expanded gene families in *T. cruzi*. [50,51,52] Recently, the TS catalytic activity proposed as a virulence factor has been confirmed and mutants lacking this activity cannot establish infection in mice.[53] Although many TS superfamily proteins do not have TS catalytic activity, TS or TS-like genes were classified altogether into eight groups.[50,51] As for other superfamilies in *T. cruzi*, there is a high variability of the member numbers of TS among strains, from ~ 800 in CL Brener to ~2300 in Bug2148.[44,45] In CL Brener, TS group I includes ~19 catalytically active TS,[50,51,54] namely TCNA and SAPA (shed acute-phase antigen) expressed in trypomastigotes, and TS-epi expressed in epimastigotes. Five genes from this group increase their expression relative to D7 [Supplementary data (Table II, Fig. 3)]. TS group II comprises a set of ~117 diverse GPI-anchored surface glycoproteins[50,51] expressed in the trypomastigote and intracellular amastigote forms of the parasites.[55,56] Proteins belonging to this group have been implicated in parasite adhesion and invasion of host cells.[56,57,58,59] Our transcriptomic analysis revealed that 46 transcripts for TS group II increased their expression relative to D7, (22 of them with significant values including 4 ERT) [Fig. 4, Supplementary data (Table II)]. Several genes coding

for TS groups, including group III , which encompasses proteins involved in the complement system (see below), and group IV to VIII, with still unknown function,[50,51] also increase their expression in the prolonged nutrient restricted stationary parasite populations comparing with the exponential parasites [Supplementary data (Table II, Fig. 3)]. Besides, many ungrouped TS coding transcripts are also upregulated in the stationary growth phase [Supplementary data (Table I)].

GP63 metalloproteases also constitute a biologically relevant cell surface family of proteins involved in trypomastigote-host cell infection.[60,61,62] Although the GP63 family is quite big, with more than 400 genes and pseudogenes, mRNAs corresponding to only 31 genes have been identified.[63] Most show significantly upregulated expression, including 3 ERT [Fig. 4, Supplementary data (Table II)].

The expression of the TcTASV family, a group of surface proteins mainly expressed in bloodstream trypomastigotes,[64,65] also shows a gradual and significant increase in the expression of some members in the long stationary phase of the epimastigote growth culture during prolonged starvation [TcCLB.509123.10, Tc-CLB.510717.10, TcCLB.510717.20, TcCLB.511877.10, Fig. 4, Supplementary data (Table II)].

Amastins constitute another group of structurally related surface proteins first identified in *T. cruzi*[66] and

then in *Leishmania*.[67] Although several roles have been assigned to amastins, their exact role in infection and disease progression is still uncertain. These proteins have been reported to constitute one of the most immunogenic surface antigens[68,69,70,71] producing strong immune responses in humans[72,73] and therefore seem to be key proteins in the host-parasite interaction. Though four groups of amastins (α-, β-, γ- and δ) have been recognised, in *T. cruzi* only the existence of β- and δ-amastins has been reported.[74] Nonetheless, the *T. cruzi* genome also bears annotated amastin genes, not assigned to β- or δ- amastin group. The expression of δ-amastins is restricted to amastigotes, whereas β-amastins are expressed in epimastigotes.[75] We found that while the expression of the epimastigote β-amastin remains almost unchanged, the δ-amastin (TcCLB.507485.150) significantly increases the expression in the long stationary phase of the epimastigote growth culture during prolonged starvation [Fig. 4, Supplementary data (Table II)].

Although, it has been reported that using proteomic approaches for the recently differentiated epimastigotes derived from trypomastigotes,[35] the upregulation of some surface proteins such as: a cruzipain protein group (TcCLB.507603.260, TcCLB.507603.270, TcCLB.509429.320 and TcCLB.509429.329), a GP63 (TcCLB.506435.370) and a trans-sialidase (TcCLB.509257.10), no significant upregulation was found for the encoding genes in the transcriptomic data here analysed.

In summary, the long stationary phase of the epimastigote growth culture during prolonged starvation exhibits some of the molecular characteristics of epimastigotes (TcSMUG and β-amastins) together with some of the metacyclic trypomastigote stage (TS, GP63 and TcTASV) and even δ-amastin being in accordance with the differentiation process but also supporting that late stationary parasites have the capacity to strongly regulate gene expression and suggesting that the intermediate parasite forms may hold cell attachment and invasion potential.

*Expression of genes involved in host complement resistance during prolonged starvation of T. cruzi epimastigote* - Considering the expression profile of surface proteins involved in the infectivity process during prolonged starvation of *T. cruzi* epimastigote, we wondered about the transcriptomic behaviour of genes coding for proteins involved in host complement bypass, one of the early mechanisms of innate immunity. The mechanisms governing complement resistance appear to be multifactorial, involving the expression of complement receptors on their surface. Parasites in their trypomastigote stage express several complement regulatory proteins[76] and/ or capture components with complement regulatory activity from the host bloodstream[77] whose molecular inhibitory mechanisms are only partially understood.[78] Among them, we focused on the calreticulin (TcCRT), the complement regulatory protein (TcCRP), and the trypomastigote decay-accelerating factor (T-DAF).

TcCRT, originally named Tc45, is a multifunctional virulence factor that participates not only in the inhibition of classical and lectin complement system activation[79,80,81] but also in the differentiation to the trypo-

mastigote form.[82] While transcriptomic analyses did not detect an increased expression of the TcCRT gene (TcCLB.509011.40) in the metacyclic trypomastigote compared to the epimastigote form,[15,83] higher expression was observed in the bloodstream trypomastigote,[83] and in the intracellular amastigote form.[84] Consistently with the reported metacyclic trypomastigote profile, we found an early drop of TcCRT gene expression that is maintained along the long stationary phase during prolonged starvation of *T. cruzi* epimastigote (Fig. 5).

TcCRP is a trypomastigote surface glycoprotein with sequence similarities to the TS family,[85] that inhibits both classical and alternative complement system activation[86] and shows a positive correlation between expression levels and strain virulence.[87] The stable transfection of *T. cruzi* epimastigotes with TcCRP-10 cDNA confers complement resistance.[88] TcCRP is encoded by the large FL-160 gene family[89] and proteins encoded by these genes share sequence similarity with members of the TS group III.[50,51] Members of this family (TcCLB.423205.10, TcCLB.504425.10, and TcCLB.511911.60) showed an increased transcript expression in the long stationary phase during prolonged starvation of *T. cruzi* epimastigote (Fig. 5).

Finally, T-DAF is an 87-93 kDa surface glycoprotein that inhibits both classical and alternative complement system activation (and probably also the lectin path-

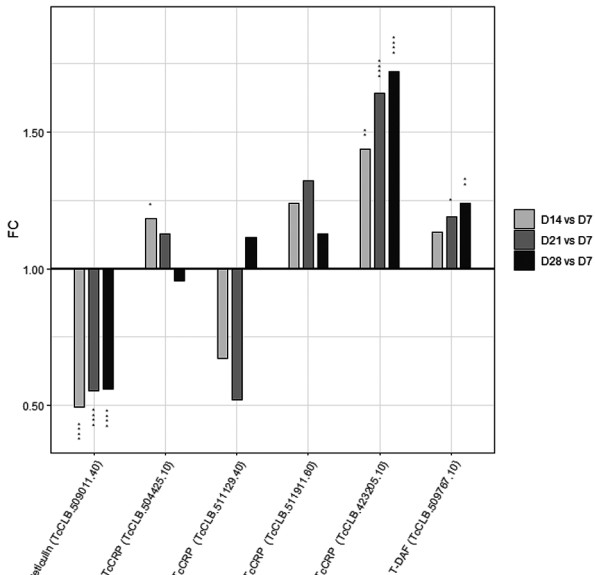

Fig. 5: expression of genes involved in complement resistance during prolonged starvation of *Trypanosoma. cruzi* epimastigote culture. The expression profile of genes of the indicated surface protein families is shown. During the prolonged starvation of *T. cruzi* epimastigote culture time points at day 7, corresponding to the exponential phase, day 14, early stationary phase, day 21, intermediate stationary phase and day 28, the final of the stationary phase (D7, D14, D21 and D28 respectively) were selected for analysis. Light grey bars represent the expression at D14 relative to D7; grey bars the expression at D21 relative to D7 and black bars the expression at D28 relative to D7. Vertical asterisks over each bar indicate adjusted significance: * p < 0.05, ** p < 0.01, *** p < 0.001.

way)[90] with higher expression (TcCLB.509767.10) in the metacyclic trypomastigote[15,83] and trypomastigote forms[84] relative to the epimastigote forms. Here, a gradual increase in the expression of this gene in the D14, D21, and D28 parasite populations was found (Fig. 5).

The increased expression of genes encoding some complement evasion proteins suggests that the intermediate parasite forms during prolonged starvation of *T. cruzi* epimastigote may be able to bypass host complement growth inhibition. In order to understand the developmental pattern of *T. cruzi* complement resistance acquisition, we studied the cell viability of the D7, D14, D21 and D28 parasite populations after treatment with complement inactivated or non-inactivated human serum by flow cytometry using propidium iodide (PI) (Fig. 6). When the parasite populations were treated with heat-inactivated complement, a low percentage of PI-positive cells, indicative of cells with disrupted or absent membranes,[91] was detected. In these conditions, maybe as a consequence of the nutrient restrictions, a slight increase in the percentage of PI labelled cells along the long stationary phase during prolonged starvation of *T. cruzi* epimastigote was observed. Conversely, after treatment with non-inactivated human serum, and in accordance with the reported complement susceptibility for *T. cruzi* epimastigotes,[92] we found a high percentage of PI-labelled cells at D7. A similar pattern is observed at D14, supporting that complement resistance mechanisms are still mostly absent at the beginning of the long stationary

phase during prolonged starvation of *T. cruzi* epimastigote. But later, in the intermediate stationary phase (D21), an increase in viable parasites is observed, leading to a parasite population mostly resistant to human complement at the final stationary phase (D28).

The increasing resistance to the host complement system of the parasite population in the long stationary phase during prolonged starvation of *T. cruzi* epimastigote culture suggests that the molecular mechanisms to evade the complement system acquired by the intermediate parasite forms are functioning.

*Autophagy genes during prolonged starvation of T. cruzi epimastigote* - Since nutritional stress conditions constitute a strong stimulus for autophagy and differentiation in *T. cruzi*, being activated during,[4,93,94] we studied the transcriptomic behaviour of genes coding for proteins involved in this process during prolonged starvation of *T. cruzi* epimastigote. Autophagy is a constitutive catabolic process responsible for self-degradation and reutilisation of the cell components, which is necessary to provide amino acids as the energy source for cell survival and to maintain cellular homeostasis, where portions of the cytoplasm are assembled in vesicles called autophagosomes that are fused with lysosomes.[92,95]

The ubiquitin-like protein Atg8 that acts on vesicle expansion and completion,[63] has two homologs in *T. cruzi*: TcAtg8.1 and TcAtg8.2.[94] We found that expression of TcAtg8.2 gene (TcCLB.510533.180) increases during the long stationary phase of prolonged starvation of *T. cruzi* epimastigote culture. Similarly, mRNAs codifying for genes involved in the Atg8 conjugation, such as Atg7 (TcCLB.507711.150) and the Atg8 processing protein Atg4 (TcCLB.511527.50), also increase their expression. A summary of the expression profile of genes related to autophagy during the prolonged starvation of *T. cruzi* epimastigote culture is shown in Fig. 7 and Supplementary data (Fig. 4).

In conclusion, to further characterise the molecular changes accompanying the nutrient restriction within the insect dwelling parasite stage, we here deep on the analysis of the reported transcriptomic database derived from the parasite population obtained along the axenic growth of *T. cruzi* epimastigotes for more than 30 days without nutrient supplementation.[36]

In the assayed conditions, we observed the known co-existence of epimastigotes, metacyclic trypomastigotes as well as an increasing proportion of intermediate parasite forms with the nucleus-kinetoplast location characteristic of epimastigote and different growth resume ability.

To analyse the transcriptomic dynamics during prolonged starvation of *T. cruzi* epimastigote culture, we here discriminate early and late response transcripts. In addition, we delved into groups of genes not addressed in our previous work, which are of great importance in the parasite life cycle. We observed a rapid change in surface protein genes such as TS and GP63. Also, the expression of the surface proteins TcTASV and δ-amastin showed upregulation. In addition, an increasing expression of genes coding for autophagy (Atg4, Atg7, Atg8.2) and complement resistance (TcCRP and T-DAF) was found, and the latter was experimentally verified.

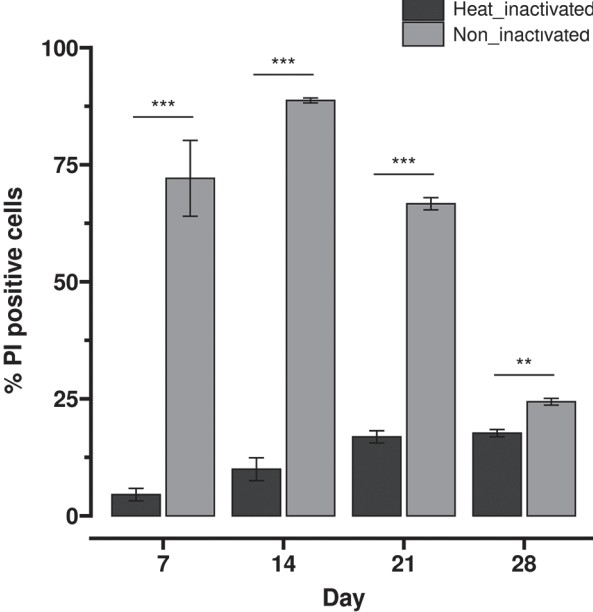

Fig. 6: complement resistance during prolonged starvation of *Trypanosoma cruzi* epimastigote culture. Percentages of PI-positive parasites representing non-viable cells of the parasite populations from the prolonged starvation of *T. cruzi* epimastigote culture at day 7 (exponential phase), day 14 (early stationary phase), day 21 (intermediate stationary phase) and day 28 (final of the stationary phase) after incubation with human serum that had or had not been heat treated. At least three independent experiments were performed for each point analysed. Significant differences are indicated: ** = p < 0.01, *** = p <0.001.

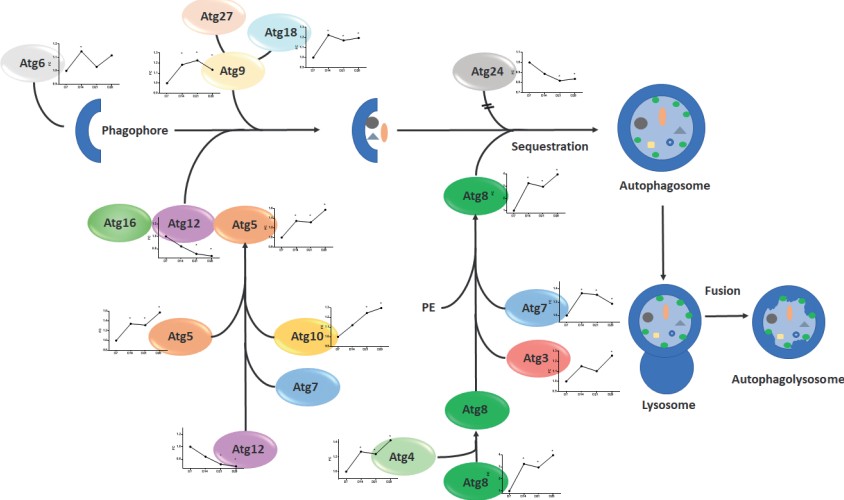

Fig. 7: expression of genes involved in autophagy identified during prolonged starvation of *Tryapnosma cruzi* epimastigote culture. Autophagy-related genes (Atg) can be grouped according to their functions at key stages of the autophagy pathway. Atg6 is involved in the initiation of the phagophore. The phagophore elongation requires the sequential activation of Atg5-Atg12-Atg16 complex). During starvation, Atg8 is cleaved by Atg4 and conjugated with phosphatidylethanolamine (PE), to then be inserted in the autophagosome membrane during its maturation, contributing to vesicle elongation. Finally, the autophagosome fusion occurs with the lysosomes. The mean of the normalised read count with its standard error is shown in each graph along the prolonged starvation of *T. cruzi* epimastigote culture at day 7, corresponding to the exponential phase, day 14, early stationary phase, day 21, intermediate stationary phase and day 28, the final of the stationary phase (D7, D14, D21 and D28 respectively). Atg6 (TcCLB.507809.119), Atg27 (TcCLB.511529.59), Atg9 (TCCLB.506925.450), Atg18 (TcCLB.509669.100), Atg12 (TcCLB.511211.104), Atg7 (TcCLB.507711.150), Atg5 (TcCLB.509965.280), Atg10 (TcCLB.507389.50), Atg24 (TCCLB.510749.30), Atg8 (TcCLB.510533.180), Atg4 (TcCLB.509443.30 and TCCLB.511527.50), Atg3 (TcCLB.510257.90), Atg7 (TCCLB.507711.150).

These results complement the analysis of processes that characterises the distinctive transcriptomic we reported for transitional parasites obtained along the axenic growth of *T. cruzi* epimastigotes for more than 30 days without nutrient supplementation,[36] supporting previous proposals of the existence of a specific parasite stage that morphologically resembles epimastigotes but exhibits distinctive biological characteristics.

## ACKNOWLEDGEMENTS

To all members of the Sección Genómica Funcional at Facultad de Ciencias, UDELAR, and the Departmento de Genómica at IIBCE for constant discussion and technical support. We also thank the colleagues that have provided critical insight into this study.

## AUTHORS' CONTRIBUTION

BG conceptualised the project, obtained funding and administered the project; LP-D, PS, MD and BG conceived specific aims and designed general strategies and experiments; LP-D designed analysis of early and late response genes and complement resistance experiments; LP-D, FH and MC performed parasite culture and quantification, RNA extraction and quantification, and complement resistance experiments; PS performed the bioinformatic analysis; LP-D, PS, MD and BG analysed the data; LP-D, PS and BG wrote the first draft of the manuscript. All authors contributed to manuscript revision, read, and approved the submitted version. The authors declare that the research was conducted in the absence of any commercial or financial relationships that could be construed as a potential conflict of interest

## DATA AVAILABILITY

A publicly available dataset was analysed in this study. This dataset can be found in the National Centre for Biotechnology (NCBI), Sequence Read Archive (SRA) BioProject ID PRJNA915394.

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

# OPEN PEER REVIEW

Memórias do IOC thanks the anonymous reviewers for their contribution to the peer review of this work.

**FIRST REVIEW ROUND**

REVIEWERS' COMMENTS

**REVIEWER #1**

Major comments:

The study complements previous findings by the group (Smircich et al., 2023) and intends to present additional evidence supporting the existence of a specific developmental stage of the parasite that, while morphologically resembling an epimastigote, displays distinct biological characteristics—particularly features associated with infectivity. However, the novelty and specific contribution of the present manuscript remain unclear.

The abstract is generally well written; however, it would benefit from a clearer and more specific articulation of the paper's main contributions. For instance, in the section corresponding to lines 28 to 40, the description remains vague with respect to the altered gene(s) or gene sets, the nature of these alterations, and how these findings differ from or advance existing knowledge in the literature. To better contextualize and underscore the relevance of the study's contributions, I recommend citing and discussing the articles I describe below, which should be incorporated into the manuscript.

Although the authors have conducted experimental assays, the notion that epimastigote forms may exhibit differing biological properties has already been proposed by other groups. For instance, Almeida-de-Faria et al. (1999) (https://doi.org/10.1006/expr.1999.4423), not cited in the manuscript, described an intracellular epimastigote form based on morphometric, biological, and biochemical parameters, as well as ultrastructural antigen localization. This form demonstrates distinct energy and carbon source requirements compared to other intracellular stages of T. cruzi(10.1016/j.molbiopara.2009.07.006, also not cited in the manuscript). Additionally, another study described epimastigote forms derived from metacyclic trypomastigotes—referred to as "recently differentiated epimastigotes"—which possess the capacity to infect mammalian cells and exhibit resistance to complement-mediated lysis (doi:10.1111/mmi.13653, not cited in the manuscript).

A review by De Souza and Barrias (2020) also discusses papers demonstrating that morphologically epimastigote-like forms, termed "transitional epimastigotes", which are capable of infecting host cells. This same review highlights findings demonstrating that T. cruzi responds to nutritional stress by repressing protein synthesis and protecting mRNA from degradation via sequestration into mRNA granules—evidence of gene expression regulation during early differentiation events.

The group itself previously identified differentially expressed genes that serve as markers of the transitional parasite population enriched under nutrient-restricted conditions, supporting the existence of a distinct developmental stage between the recognized insect-dwelling forms.

It is important in the presente manuscript to explicitly the genes or gene groups (markers) that meaningfully expand upon previously analyzed data, particularly in the context of nutritional stress response (long starvation). If the aim is to experimentally validate transcriptomic observations and thereby confirm findings derived from in silico analyses, this needs to be clearly distinguished from earlier results. Not only from those mentioned above, but also from classical studies which have already shown that stage-specific gene activation precedes by several days the morphological changes observed during the epimastigote-to-trypomastigote transformation in T. cruzi (e.g., Goldenberg, 1985: https://doi.org/10.1016/0166-6851(85)90108-2; https://doi.org/10.1016/0014-5793(85)81083-8—both not cited in the manuscript).

Hernandez's work on stationary-phase T. cruzi epimastigotes has already described morphological and metabolic changes as consequences of nutritional stress. Similarly, studies on metacyclogenesis have established nutritional stress as a key trigger for parasite differentiation (e.g., https://doi.org/10.1016/0166-6851(85)90073-8, not cited). Morphological studies further demonstrate gradual transitions during metacyclogenesis, based on morphological and ultrastructural analyses of differentiating cells (https://doi.org/10.1186/s13071-018-2664-4). In addition, inhibition of the ubiquitin-proteasome pathway in T. cruzi epimastigotes has been shown to disrupt cell division and differentiation processes, affecting nuclear and kinetoplast DNA distribution (https://doi.org/10.1007/s00436-008-1081-6, also not cited).

Based on mentioned above, I strongly encourage the authors to revise the manuscript accordingly, or even reconsider to reanalise the data, in light of the studies cited here, which were neither referenced nor discussed. While the manuscript does highlight and discuss gene groups associated with complement resistance, autophagy, and known surface pathogenic factors, these aspects have already been somehow documented in the context of parasite stage differentiation. Maybe the description of genes would be the diferential, but somehow it should be more associated to the biological relevance of the prolonged starvation phase within the T. cruzi life cycle,

because it is not clearly addressed. Beyond the microscopy and complement lysis assay, no additional functional experiments were performed. Therefore, it is critical—based on the data presented—to clearly articulate how this work advances the field beyond what has already been published.

Minor comments:
• Line 33, page 10 - "some metacyclic transcript markers". Should specify which markers.
• Line 6, page 13 -  considering discussing in the light of endocytic ability, see  the review paper : 10.1016/j.jsb.2016.07.018.
• Line 37, page 19:  T.cruzi should be in italic.
• Line 17, page 26 - I suggest to adequate the phrase to "When parasite populations were treated with heat-inactivated complement, a low percentage of PI-positive cells—indicative of cells with disrupted or absent membranes [77]—was detected."
• Line 38, page 29: "These results complement the distinctive transcriptomic we reported for transitional parasites obtained along the axenic growth of T. cruzi epimastigotes for more than 30 days without nutrient supplementation [25]." I thinks it is still vague, and the authors should more specific in which genes or which data are the complement in comparison to the previus work.
• Sfigure1 and Figure 7 are too small.

**REVIEWER #2**

In this work, Perez-Dias et al described a transcriptome analysis based on data that was previously generated by the same group, in which global gene expression of T. cruzi epimastigotes kept in culture for more than 30 days without nutrient supplementation (prolonged starvation) was examined. Distinct from the previously reported analysis, which focused on genes related to protein quality or content, metabolic switch from glucose to amino acid consumption, redox challenge, and surface proteins expression, in this report, the authors focused on other processes related to epimastigotes/metacyclic trypomastigotes differenctiation: parasite autophagy and complement resistance. This is a relevant study that investigates a key process during the T. cruzi life cycle, which contributes to the molecular understanding of parasite differentiation and propagation in the insect vector.

The abstract is adequate but could be improved with minor changes that would more faithfully describe the analyses that were performed. As a suggestion, the sentence "Since nutrient restriction is considered one of the main factors driving metacyclogenesis and is very frequent due to the long-term starvation periods that the insect vectors commonly undergo, we have studied the transcriptomic effects of nutrient restriction on long-lasting epimastigote cultures.", could be changed to "Since nutrient restriction is considered one of the main factors driving metacyclogenesis and is very frequently due to the long-term starvation periods that the insect vectors commonly undergo, we have studied the morphological and growth rate alterations as well as transcriptomic changes that result from nutrient restriction on long-lasting epimastigote cultures.

In the Introduction (on page 5), I suggest including a more complete description of the literature regarding RNA-seq analyses published by other groups (please see García-Huertas et al., 2023, Transcriptional changes during metacyclogenesis of a Colombian Trypanosoma cruzi strain. doi: 10.1007/s00436-022-07766-3) as well as a few reports describing the role of RNA binding proteins involved with gene expression control during metacyclogenesis (please see Tavares et al., 2021, Trypanosoma cruzi zinc finger protein that is implicated in the control of epimastigote-specific gene expression and metacyclogenesis. doi: 10.1017/S0031182020002176; Alcantara et al., 2018, Knockout of the CCCH zinc finger protein TcZC3H31 blocks Trypanosoma cruzi differentiation into the infective metacyclic form. doi: 10.1016/j.molbiopara.2018.01.006).

In the Material and Methods section, although the authors referred to a previous published study, important information regarding the parasite strain, and RNA-seq analyses, should be included.

In the conclusion section, I suggest removing the last sentence: "In addition, these findings reinforce previous warnings and mistrust the well accepted nonpathogenic characteristic of the epimastigote stage".

I suggest several minor changes to be made in the Results/Discussion section:

1- Please add in Figure 1 a quantification of the distinct parasite populations based on the morphology analysis, with the repositioning of the kinetoplast as a criterion to define the transformation into metacyclic forms. This is an important point since, on page 16, the authors indicate that the expression of metacyclic markers "remain constant all along the prolonged starvation of the T. cruzi  epimastigote culture suggesting that metacyclic trypomastigotes are not significatively contributing to transcriptomic data". To better discuss this important point, it would be essential do quantify the % of metacyclic forms.

As an alternative, the authors should discuss the possibility that Metacyclin II and III are genes that may be not necessarily good markers for differentiation, since their products have not been characterized.

2- (page 12), please describe more clearly the different combinations of data sets for the comparative analysis of DEGs between the selected time points. Also, it is not clear the meaning of the numbers above the graphs on SFigure2.

3- Data from SFigure 1 did not indicate an increased doubling time for the D28 parasite population, as stated on page 16. Also, SFigure1 should be cited before SFigure 2 in the text.

4- On page 19, the authors wrongly state that among surface proteins, mucins and MASPs constitute the most expressed gene families in T. cruzi. However, the largest gene family if Trans-sialidase and TS genes are not only highly expressed in trypomastigotes, but it is also expressed in other parasite stages. This information, added in the following paragraph and should be included in the beginning of this section. Also, the data related to copy numbers of members of TS family encoding active enzymes should be based on other studies (please see Wang et al., 2021, Strain-specific genome evolution in Trypanosoma cruzi, the agent of Chagas disease. doi:10.1371/journal.ppat.1009254; Burle-Caldas et al., 2022, Disruption of active trans-sialidase genes impairs egress from mammalian host cells and generates highly attenuated Trypanosoma cruzi, doi.org/10.1128/mBio.03246-19). Plus, it has been recently shown that, different from what was suggested, TS with enzymatic activity, are not involved in parasite adhesion and invasion of host cells but is mainly involved in parasite egress from the infected cell and survival in the bloodstream of the mammalian host (see Burle-Caldas et al., 2022).

5- On Figure 4, the observation that increased expression, on day 28, of δ-amastins and TcTASV with FC > 10 could be emphasized, not only because it is in accordance with the differentiation to the infective, mammalian stage, but also because it shows that late stationary parasites have the capacity to strongly up-regulate gene expression.

6- Incorrect read mapping may affect the results shown in SFigure 3, since it is well known that short Illumina reads are difficult to map precisely into the different members of TS family, the authors should discuss this possibility. In addition to imprecise mapping, the small values for FC for TS group I and III may indicate that changes in the expression of these genes may not be biologically relevant.

7- Regarding data shown in Figure 6, it is not clear what the author means by this sentence: The presence of the nonreplicative and complement resistant metacyclic trypomastigotes cannot completely explain the huge number of viable cells observed.

8- Data from Figure 7 is impossible to analyze, so please, show larger graphs with numbers of FC that can be read.

## AUTHORS' RESPONSE TO THE REVIEWERS

Dr. Adeilton Brandão
Handling Editor
Memórias do Instituto Oswaldo Cruz

We are hereby submitting a new revised version of the manuscript. All the reviewer´s concerns were attended and the responses to each of them is presented below. We have agreed with most of the recommendations, acknowledged the suggestions and incorporated them in this new version.

Figure 7 and SFigure 1 were modified as suggested. Besides, Figure 4 and SFigure 3 were improved and the data corresponding to these figures was added also as a new supplementary table (STable II) to highlight the coordinated modulation of multigene families associated with parasite adaptation and persistence within the host. Figure 5 was also redesigned. In addition, a new supplementary figure was added to clarify one concern of both reviewers (SFigure 4).

We would like to thank all for the reviewer process, which has undoubtedly contributed to improving the manuscript and hope you consider this new version suitable for publication

Best regards
Dr. Beatriz Garat

REVIEWER COMMENTS:
Reviewer: 1
Reviewer comments:
Major comments:
The study complements previous findings by the group (Smircich et al., 2023) and intends to present additional evidence supporting the existence of a specific developmental stage of the parasite that, while morphologically resembling an epimastigote, displays distinct biological characteristics—particularly features associated with infectivity. However, the novelty and specific contribution of the present manuscript remain unclear.

The abstract is generally well written; however, it would benefit from a clearer and more specific articulation of the paper's main contributions. For instance, in the section corresponding to lines 28 to 40, the description remains vague with respect to the altered gene(s) or gene sets, the nature of these alterations, and how these findings differ from or advance existing knowledge in the literature.

R: The text was modified to include some of the specific genes within the groups:

"We found a gene expression early rise of surface protein (such as TS and GP63) and even a rise of TcTASV and δ-amastin, which is not accompanied by increased expression of metacyclic transcript markers. In addition,

we found increased expression of genes coding for proteins involved in two other processes activated during the differentiation of epimastigotes to the infective form of the parasite: autophagy (Atg4, Atg7, Atg8.2) and complement resistance (TcCRP and T-DAF)."

To better contextualize and underscore the relevance of the study's contributions, I recommend citing and discussing the articles I describe below, which should be incorporated into the manuscript. Although the authors have conducted experimental assays, the notion that epimastigote forms may exhibit differing biological properties has already been proposed by other groups.

R: We certainly agree with the reviewer and consider that we have already clarified this point in the manuscript. Indeed, we conclude that the results we provide here constitute new evidence to support these previous proposals.

For instance, Almeida-de-Faria et al. (1999) (https://doi.org/10.1006/expr.1999.4423), not cited in the manuscript, described an intracellular epimastigote form based on morphometric, biological, and biochemical parameters, as well as ultrastructural antigen localization. This form demonstrates distinct energy and carbon source requirements compared to other intracellular stages of T. cruzi (10.1016/j.molbiopara.2009.07.006, also not cited in the manuscript.

R: Since the work presented in the manuscript is restricted to the insect-dwelling part of the cycle, we have decided not to include the discussion of the papers that distinctly focus on the mammalian part of the life cycle, particularly on an intracellular form. Nonetheless, we have now included a brief comment addressing the reviewer's concern with the corresponding references. The following text was included:

"The plasticity and complexity of T. cruzi forms along the life cycle spread beyond the epimastigote to the metacyclic trypomastigote transition. Transient T. cruzi epimastigote-like forms as intermediates in the differentiation of amastigotes to trypomastigotes inside the mammalian host cells [32] and their distinct energy and carbon source requirements compared to the other intracellular stages [33] have been characterized."

Additionally, another study described epimastigote forms derived from metacyclic trypomastigotes—referred to as "recently differentiated epimastigotes"—which possess the capacity to infect mammalian cells and exhibit resistance to complement-mediated lysis (doi:10.1111/mmi.13653, not cited in the manuscript).

R: We thank the reviewer for the recommendation to include this paper. Using cell biology, this paper characterizes the infection ability of recently differentiated epimastigotes obtained from cell derived or metacyclic trypomastigotes. It analyzes, using shotgun, the protein expression of virulence factors of recently differentiated epimastigotes obtained from cell derived trypomastigotes. We have now included this reference in the Introduction section:

"In addition, the differentiation from the trypomastigote forms (cell-derived or metacyclic) to an epimastigote-like form named "recently differentiated epimastigotes", exhibiting complement resistance and infection ability has been recently described using cell biology and proteomic approaches [34]."

The comparison of the proteomic approach of recently differentiated epimastigotes obtained from cell derived trypomastigotes with the transcriptomic data during prolonged starvation of Trypanosoma cruzi epimastigote related to the acquisition of infective and complement resistance was included in the Results section.

"Although, it has been reported that using proteomic approaches for the recently differentiated epimastigotes derived from trypomastigotes [34], the upregulation of some surface proteins such as: a cruzipain protein group (TcCLB.507603.260, TcCLB.507603.270, TcCLB.509429.320 and TcCLB.509429.329), a GP63 (TcCLB.506435.370) and a trans-sialidase (TcCLB.509257.10), no significant differential expression was found for the encoding genes in the transcriptomic data here analyzed."

A review by De Souza and Barrias (2020) also discusses papers demonstrating that morphologically epimastigote-like forms, termed "transitional epimastigotes", which are capable of infecting host cells. This same review highlights findings demonstrating that T. cruzi responds to nutritional stress by repressing protein synthesis and protecting mRNA from degradation via sequestration into mRNA granules—evidence of gene expression regulation during early differentiation events. The group itself previously identified differentially expressed genes that serve as markers of the transitional parasite population enriched under nutrient-restricted conditions, supporting the existence of a distinct developmental stage between the recognized insect-dwelling forms. It is important in the presente manuscript to explicitly the genes or gene groups (markers) that meaningfully expand upon previously analyzed data, particularly in the context of nutritional stress response (long starvation). If the aim is to experimentally validate transcriptomic observations and thereby confirm findings derived from in silico analyses, this needs to be clearly distinguished from earlier results.

R: Both papers were presented in the manuscript. Distinct from De Souza and Barrias, as stated in the manuscript, we use transcriptomic approaches to characterize the differential expression of epimastigotes upon long starvation. Distinctly from our previous paper, as stated in the manuscript, we here focus on genes coding for surface proteins and proteins related to complement resistance and autophagy.

Not only from those mentioned above, but also from classical studies which have already shown that stage-specific gene activation precedes by several days the morphological changes observed during the epimastigote-to-trypomastigote transformation in T. cruzi (e.g., Goldenberg, 1985: https://doi.org/10.1016/0166-6851(85)90108-2; https://doi.org/10.1016/0014-5793(85)81083-8—both not cited in the manuscript). Hernandez's work on stationary-phase T. cruzi epimastigotes has already described morphological and metabolic changes as consequences of

nutritional stress. Similarly, studies on metacyclogenesis have established nutritional stress as a key trigger for parasite differentiation (e.g., https://doi.org/10.1016/0166-6851(85)90073-8, not cited). Morphological studies further demonstrate gradual transitions during metacyclogenesis, based on morphological and ultrastructural analyses of differentiating cells (https://doi.org/10.1186/s13071-018-2664-4). In addition, inhibition of the ubiquitin-proteasome pathway in T. cruzi epimastigotes has been shown to disrupt cell division and differentiation processes, affecting nuclear and kinetoplast DNA distribution (https://doi.org/10.1007/s00436-008-1081-6, also not cited).

R: We have now included the references recommended by the reviewer. These include papers of recognized relevance that constitute basic contributions and/or focus on parasite developmental transitions in different conditions. Globally, all of them together with the results we are presenting certainly contribute to obtaining a more integral landscape of the complex changes the parasite may undergo in the insect vector.

Based on mentioned above, I strongly encourage the authors to revise the manuscript accordingly, or even reconsider to reanalise the data, in light of the studies cited here, which were neither referenced nor discussed. While the manuscript does highlight and discuss gene groups associated with complement resistance, autophagy, and known surface pathogenic factors, these aspects have already been somehow documented in the context of parasite stage differentiation. Maybe the description of genes would be the diferential, but somehow it should be more associated to the biological relevance of the prolonged starvation phase within the T. cruzi life cycle, because it is not clearly addressed. Beyond the microscopy and complement lysis assay, no additional functional experiments were performed. Therefore, it is critical—based on the data presented—to clearly articulate how this work advances the field beyond what has already been published.

R: Following the reviewer's advice the text was modified to highlight the novelty of the here presented dynamic analysis of transcriptomic changes along prolonged starvation of epimastigotes searching for evidence in our result that could lead to proposing similarities or differences with the published data using different approaches or conditions of metacyclogenesis and/or epimastigogenesis. Following the reviewer's recommendation, we have also highlighted the behavior of specific genes within the analyzed groups.

Minor comments:
• Line 33, page 10 - "some metacyclic transcript markers". Should specify which markers.
R: Text was modified to specify the markers:
"some metacyclic transcript markers such as metacyclin II and III."
• Line 6, page 13 - considering discussing in the light of endocytic ability, see the review paper: 10.1016/j.jsb.2016.07.018.
R: Text was modified to include the mentioned reference:
"Consequently, repositioning of the kinetoplast and loss of endocytic ability have only been observed in the later stages of the metacyclogenesis process [5, 10.1016/j.jsb.2016.07.018]"
• Line 37, page 19: T.cruzi should be in italic.
R: Text was corrected
• Line 17, page 26 - I suggest to adequate the phrase to "When parasite populations were treated with heat-inactivated complement, a low percentage of PI-positive cells—indicative of cells with disrupted or absent membranes [77]—was detected."
R: Text was modified accordingly
• Line 38, page 29: "These results complement the distinctive transcriptomic we reported for transitional parasites obtained along the axenic growth of T. cruzi epimastigotes for more than 30 days without nutrient supplementation [25]." I thinks it is still vague, and the authors should more specific in which genes or which data are the complement in comparison to the previus work.
R: Following the reviewer′s observation, the text was modified as follows:
"To analyze the transcriptomic dynamics during prolonged starvation of T. cruzi epimastigote culture, we here discriminate early and late response transcripts. In addition, we delved into groups of genes not addressed in our previous work, which are of great importance in the parasite life cycle. We observed a rapid change in surface protein genes such as TS and GP63. Also, the expression of the surface proteins TcTASV and δ-amastin showed upregulation. In addition, an increasing expression of genes coding for autophagy (Atg4, Atg7, Atg8.2) and complement resistance (TcCRP and T-DAF) was found, and the latter was experimentally verified.

These results complement the analysis of processes that characterizes the distinctive transcriptomic we reported for transitional parasites obtained along the axenic growth of T. cruzi epimastigotes for more than 30 days without nutrient supplementation…"
• Sfigure1 and Figure 7 are too small.
R: Thanks for the comment. We have enlarged the indicated figures.

Reviewer: 2
Reviewer comments:
In this work, Perez-Dias et al described a transcriptome analysis based on data that was previously generated by the same group, in which global gene expression of T. cruzi epimastigotes kept in culture for more than 30 days

without nutrient supplementation (prolonged starvation) was examined. Distinct from the previously reported analysis, which focused on genes related to protein quality or content, metabolic switch from glucose to amino acid consumption, redox challenge, and surface proteins expression, in this report, the authors focused on other processes related to epimastigotes/metacyclic trypomastigotes differenctiation: parasite autophagy and complement resistance. This is a relevant study that investigates a key process during the T. cruzi life cycle, which contributes to the molecular understanding of parasite differentiation and propagation in the insect vector.

1) The abstract is adequate but could be improved with minor changes that would more faithfully describe the analyses that were performed. As a suggestion, the sentence "Since nutrient restriction is considered one of the main factors driving metacyclogenesis and is very frequent due to the long-term starvation periods that the insect vectors commonly undergo, we have studied the transcriptomic effects of nutrient restriction on long-lasting epimastigote cultures.", could be changed to "Since nutrient restriction is considered one of the main factors driving metacyclogenesis and is very frequently due to the long-term starvation periods that the insect vectors commonly undergo, we have studied the morphological and growth rate alterations as well as transcriptomic changes that result from nutrient restriction on long-lasting epimastigote cultures.

R: The text was modified following the reviewer´s suggestion.

2) In the Introduction (on page 5), I suggest including a more complete description of the literature regarding RNA-seq analyses published by other groups (please see García-Huertas et al., 2023, Transcriptional changes during metacyclogenesis of a Colombian Trypanosoma cruzi strain. doi: 10.1007/s00436-022-07766-3) as well as a few reports describing the role of RNA binding proteins involved with gene expression control during metacyclogenesis (please see Tavares et al., 2021, Trypanosoma cruzi zinc finger protein that is implicated in the control of epimastigote-specific gene expression and metacyclogenesis. doi: 10.1017/S0031182020002176; Alcantara et al., 2018, Knockout of the CCCH zinc finger protein TcZC3H31 blocks Trypanosoma cruzi differentiation into the infective metacyclic form. doi: 10.1016/j.molbiopara.2018.01.006).

R: We thank the reviewer for this observation. The citation of these works was incorporated.

3) In the Material and Methods section, although the authors referred to a previous published study, important information regarding the parasite strain, and RNA-seq analyses, should be included.

R: The information on parasite strain T. cruzi Dm28c strain (TcI DTU) was added to the Material and Methods section. We consider that the Materials and Methods section in the manuscript correctly presented the method of analysis of the RNA-seq data from the published paper (open-access) and should not include the method to obtain them (because it is a merit of the published paper).

4) In the conclusion section, I suggest removing the last sentence: "In addition, these findings reinforce previous warnings and mistrust the well accepted nonpathogenic characteristic of the epimastigote stage".

R: The text was modified following the reviewer´s suggestion.

I suggest several minor changes to be made in the Results/Discussion section:

1- Please add in Figure 1 a quantification of the distinct parasite populations based on the morphology analysis, with the repositioning of the kinetoplast as a criterion to define the transformation into metacyclic forms.

R: Given the characteristics of the transitional forms (displaying a positioning of the kinetoplast, flagellum, and nucleus corresponding to the epimastigote stage but with non-classical epimastigote morphology of the cell body and flagellum) they cannot be precisely quantified by visual inspection. Therefore, as reported in our previous paper (Smircich et al 2023), only total and metacyclic trypomastigote forms were quantified. Following the reviewer´s concern the text was modified to incorporate the quantification of this parasite form:

"In these conditions, we have found that the percentage of metacyclic trypomastigote within this long stationary phase displays a composed profile including a gradual increase, not surpassing 10% for a long period (3.8 ± 1.7%, 5.4± 0.4% and 7.9 ± 0.6% for the exponential phase -day 7-, early stationary phase -day 14-, and intermediate stationary phase-day 21 respectively) and then a sharp increase at the end (32.1 ± 5.4% for the final of the stationary phase -day 28-) [35]."

This is an important point since, on page 16, the authors indicate that the expression of metacyclic markers "remain constant all along the prolonged starvation of the T. cruzi epimastigote culture suggesting that metacyclic trypomastigotes are not significatively contributing to transcriptomic data". To better discuss this important point, it would be essential do quantify the % of metacyclic forms.

As an alternative, the authors should discuss the possibility that Metacyclin II and III are genes that may be not necessarily good markers for differentiation, since their products have not been characterized.

R: We deeply acknowledge and certainly agree with the reviewer´s observation. The text was modified as follows:

"It is worth noting that the profiles of previously reported markers of the metacyclic trypomastigotes, genes coding for Metacyclin II (TcCLB.506529.600) and Metacyclin III (TcCLB.509251.6) [6,27] remain constant all along the prolonged starvation of the T. cruzi epimastigote culture not accompanying the slow increase of metacyclic forms described above (SFigure 2)."

2- (page 12), please describe more clearly the different combinations of data sets for the comparative analysis of DEGs between the selected time points. Also, it is not clear the meaning of the numbers above the graphs on SFigure2.

R: Following the reviewer′s comment we modified the text as follows:

"In order to distinguish temporal processes triggered by prolonged starvation within the long stationary phase of T. cruzi epimastigote culture, DEGs were classified as nutrient restriction early response transcripts (ERT) if significant different expression between the data derived from epimastigotes in the exponential (D7) and in the early stationary phase (D14) was observed (subclassified in Figure 2 as following: D14 vs D7, D14 vs D7 plus D21 vs D7, D14 vs D7 and D21 vs D7 and D28 vs D7), likewise, DEGs were classified as late response transcripts (LRT) if the different expression was restricted to the changes between intermediate and late stationary phase (D28 vs D21) (STable I, SFigure 2)."

3- Data from SFigure 1 did not indicate an increased doubling time for the D28 parasite population, as stated on page 16.

R: We agree and thank the reviewer for pointing out this mistake. It should have been decreased instead of increased. The text was modified accordingly.

Also, SFigure1 should be cited before SFigure 2 in the text.

R: SFigure1 is already cited (starting at page 9 l32) before SFigure2 (starting at page 12 l40) in the manuscript. No changes were made.

4- On page 19, the authors wrongly state that among surface proteins, mucins and MASPs constitute the most expressed gene families in T. cruzi. However, the largest gene family if Trans-sialidase and TS genes are not only highly expressed in trypomastigotes, but it is also expressed in other parasite stages. This information, added in the following paragraph and should be included in the beginning of this section.

R: Following the reviewer′s observation the text for mucins and MASPs was changed:

"…mucins and MASPs constitute a highly expressed gene family in T. cruzi …"

In addition, we added "highly expressed" at the beginning of the TS superfamily paragraph

Also, the data related to copy numbers of members of TS family encoding active enzymes should be based on other studies (please see Wang et al., 2021, Strain-specific genome evolution in Trypanosoma cruzi, the agent of Chagas disease. doi:10.1371/journal.ppat.1009254; Burle-Caldas et al., 2022, Disruption of active trans-sialidase genes impairs egress from mammalian host cells and generates highly attenuated Trypanosoma cruzi, doi.org/10.1128/mBio.03246-19). Plus, it has been recently shown that, different from what was suggested, TS with enzymatic activity, are not involved in parasite adhesion and invasion of host cells but is mainly involved in parasite egress from the infected cell and survival in the bloodstream of the mammalian host (see Burle-Caldas et al., 2022).

R: Following the reviewer′s observation, both references were included

"Surface proteins, arranged as large gene families exhibiting diversification and copy number diversity [39], play different roles in the parasite's life cycle progression, in the host-cell interplay, immune system evasion and persistence of the parasite [40]."

"Recently, the TS catalytic activity proposed as a virulence factor has been confirmed and mutants lacking this activity cannot establish infection in mice [52]."

5- On Figure 4, the observation that increased expression, on day 28, of δ-amastins and TcTASV with FC > 10 could be emphasized, not only because it is in accordance with the differentiation to the infective, mammalian stage, but also because it shows that late stationary parasites have the capacity to strongly up-regulate gene expression.

R: We appreciate the suggestion. Following the reviewer′s comment the text was modified:

"In summary, the long stationary phase of the epimastigote growth culture during prolonged starvation exhibits some of the molecular characteristics of epimastigotes (TcSMUG and β-amastins) together with some of the metacyclic trypomastigote stage (TS, GP63 and TcTASV) and even δ-amastin being in accordance with the differentiation process but also supporting that late stationary parasites have the capacity to strongly regulate gene expression and suggesting that the intermediate parasite forms may hold cell attachment and invasion potential."

6- Incorrect read mapping may affect the results shown in SFigure 3, since it is well known that short Illumina reads are difficult to map precisely into the different members of TS family, the authors should discuss this possibility. In addition to imprecise mapping, the small values for FC for TS group I and III may indicate that changes in the expression of these genes may not be biologically relevant.

R: We acknowledge the challenges in accurately mapping reads to highly similar members of large surface protein families. However, these potential biases are consistent across replicates and conditions, ensuring that differential expression analysis remains robust, since any deviation from the true read count would affect both conditions proportionally.

In addition, both Figure 4 and SFigure 3 were redesigned and a new supplementary table (STable II) with data corresponding to Figures 4 and SFigure3 was added to highlight the coordinated modulation of multigene families associated with parasite adaptation and persistence within the host.

7- Regarding data shown in Figure 6, it is not clear what the author means by this sentence: The presence of the nonreplicative and complement resistant metacyclic trypomastigotes cannot completely explain the huge number of viable cells observed.

R: The sentence was removed

8- Data from Figure 7 is impossible to analyze, so please, show larger graphs with numbers of FC that can be read.

R: We appreciate the reviewer's observation regarding Figure 7. We agree that the graphs were too small to analyze properly. In the revised manuscript, we have increased the size of the panels in the central figure to improve readability. Because our aim with Figure 7 is to illustrate overall trends in gene expression across the pathway, we have also added a supplementary figure (SFigure 4), where the plots are shown at a larger scale. For clarity, each time point's fold-change (FC) relative to day 7 is indicated on the graph, and statistically significant differences vs. day 7 are denoted with an asterisk.

## SECOND REVIEW ROUND

REVIEWERS' COMMENTS

### REVIEWER #1

I am satisfied with the revised version. The modifications were appropriately incorporated and contributed substantially to improving the manuscript.

### REVIEWER #2

No comments.

