## [Reviewer Report · FIRST REVIEW ROUND - REVIEWERS COMMENTS]

## REVIEWER #1

**Major comments:**

The study complements previous findings by the group (Smircich et al., 2023) and intends to present additional evidence supporting the existence of a specific developmental stage of the parasite that, while morphologically resembling an epimastigote, displays distinct biological characteristics—particularly features associated with infectivity. However, the novelty and specific contribution of the present manuscript remain unclear.

The abstract is generally well written; however, it would benefit from a clearer and more specific articulation of the paper’s main contributions. For instance, in the section corresponding to lines 28 to 40, the description remains vague with respect to the altered gene(s) or gene sets, the nature of these alterations, and how these findings differ from or advance existing knowledge in the literature. To better contextualize and underscore the relevance of the study’s contributions, I recommend citing and discussing the articles I describe below, which should be incorporated into the manuscript.

Although the authors have conducted experimental assays, the notion that epimastigote forms may exhibit differing biological properties has already been proposed by other groups. For instance, Almeida-de-Faria et al. (1999) (*https://doi.org/10.1006/expr.1999.4423*), not cited in the manuscript, described an intracellular epimastigote form based on morphometric, biological, and biochemical parameters, as well as ultrastructural antigen localization. This form demonstrates distinct energy and carbon source requirements compared to other intracellular stages of *T. cruzi* (*10.1016/j.molbiopara.2009.07.006*, also not cited in the manuscript). Additionally, another study described epimastigote forms derived from metacyclic trypomastigotes—referred to as “recently differentiated epimastigotes”—which possess the capacity to infect mammalian cells and exhibit resistance to complement-mediated lysis (*doi:10.1111/mmi.13653*, not cited in the manuscript).

A review by De Souza and Barrias (2020) also discusses papers demonstrating that morphologically epimastigote-like forms, termed “transitional epimastigotes”, which are capable of infecting host cells. This same review highlights findings demonstrating that *T. cruzi* responds to nutritional stress by repressing protein synthesis and protecting mRNA from degradation via sequestration into mRNA granules—evidence of gene expression regulation during early differentiation events.

The group itself previously identified differentially expressed genes that serve as markers of the transitional parasite population enriched under nutrient-restricted conditions, supporting the existence of a distinct developmental stage between the recognized insect-dwelling forms.

It is important in the presente manuscript to explicitly the genes or gene groups (markers) that meaningfully expand upon previously analyzed data, particularly in the context of nutritional stress response (long starvation). If the aim is to experimentally validate transcriptomic observations and thereby confirm findings derived from in silico analyses, this needs to be clearly distinguished from earlier results. Not only from those mentioned above, but also from classical studies which have already shown that stage-specific gene activation precedes by several days the morphological changes observed during the epimastigote-to-trypomastigote transformation in *T. cruzi* (e.g., Goldenberg, 1985: *https://doi.org/10.1016/0166-6851(85)90108-2*; *https://doi.org/10.1016/0014-5793(85)81083-8*—both not cited in the manuscript).

Hernandez’s work on stationary-phase *T. cruzi* epimastigotes has already described morphological and metabolic changes as consequences of nutritional stress. Similarly, studies on metacyclogenesis have established nutritional stress as a key trigger for parasite differentiation (e.g., *https://doi.org/10.1016/0166-6851(85)90073-8*, not cited). Morphological studies further demonstrate gradual transitions during metacyclogenesis, based on morphological and ultrastructural analyses of differentiating cells (*https://doi.org/10.1186/s13071-018-2664-4*). In addition, inhibition of the ubiquitin-proteasome pathway in *T. cruzi* epimastigotes has been shown to disrupt cell division and differentiation processes, affecting nuclear and kinetoplast DNA distribution (*https://doi.org/10.1007/s00436-008-1081-6*, also not cited).

Based on mentioned above, I strongly encourage the authors to revise the manuscript accordingly, or even reconsider to reanalise the data, in light of the studies cited here, which were neither referenced nor discussed. While the manuscript does highlight and discuss gene groups associated with complement resistance, autophagy, and known surface pathogenic factors, these aspects have already been somehow documented in the context of parasite stage differentiation. Maybe the description of genes would be the diferential, but somehow it should be more associated to the biological relevance of the prolonged starvation phase within the *T. cruzi* life cycle, because it is not clearly addressed. Beyond the microscopy and complement lysis assay, no additional functional experiments were performed. Therefore, it is critical—based on the data presented—to clearly articulate how this work advances the field beyond what has already been published.

**Minor comments:**

- Line 33, page 10 - “some metacyclic transcript markers”. Should specify which markers.

- Line 6, page 13 - considering discussing in the light of endocytic ability, see the review paper : *10.1016/j.jsb.2016.07.018*.

- Line 37, page 19: *T.cruzi* should be in italic.

- Line 17, page 26 - I suggest to adequate the phrase to “When parasite populations were treated with heat-inactivated complement, a low percentage of PI-positive cells—indicative of cells with disrupted or absent membranes [77]—was detected.”

- Line 38, page 29: “These results complement the distinctive transcriptomic we reported for transitional parasites obtained along the axenic growth of *T. cruzi* epimastigotes for more than 30 days without nutrient supplementation [25].” I thinks it is still vague, and the authors should more specific in which genes or which data are the complement in comparison to the previus work.

- Sfigure1 and Figure 7 are too small.

## REVIEWER #2

In this work, Perez-Dias et al described a transcriptome analysis based on data that was previously generated by the same group, in which global gene expression of *T. cruzi* epimastigotes kept in culture for more than 30 days without nutrient supplementation (prolonged starvation) was examined. Distinct from the previously reported analysis, which focused on genes related to protein quality or content, metabolic switch from glucose to amino acid consumption, redox challenge, and surface proteins expression, in this report, the authors focused on other processes related to epimastigotes/metacyclic trypomastigotes differenctiation: parasite autophagy and complement resistance. This is a relevant study that investigates a key process during the *T. cruzi* life cycle, which contributes to the molecular understanding of parasite differentiation and propagation in the insect vector.

The abstract is adequate but could be improved with minor changes that would more faithfully describe the analyses that were performed. As a suggestion, the sentence “Since nutrient restriction is considered one of the main factors driving metacyclogenesis and is very frequent due to the long-term starvation periods that the insect vectors commonly undergo, we have studied the transcriptomic effects of nutrient restriction on long-lasting epimastigote cultures.”, could be changed to “Since nutrient restriction is considered one of the main factors driving metacyclogenesis and is very frequently due to the long-term starvation periods that the insect vectors commonly undergo, we have studied the morphological and growth rate alterations as well as transcriptomic changes that result from nutrient restriction on long-lasting epimastigote cultures.

In the Introduction (on page 5), I suggest including a more complete description of the literature regarding RNA-seq analyses published by other groups (please see García-Huertas et al., 2023, Transcriptional changes during metacyclogenesis of a Colombian Trypanosoma cruzi strain. doi: 10.1007/s00436-022-07766-3) as well as a few reports describing the role of RNA binding proteins involved with gene expression control during metacyclogenesis (please see Tavares et al., 2021, Trypanosoma cruzi zinc finger protein that is implicated in the control of epimastigote-specific gene expression and metacyclogenesis. doi: 10.1017/S0031182020002176; Alcantara et al., 2018, Knockout of the CCCH zinc finger protein TcZC3H31 blocks Trypanosoma cruzi differentiation into the infective metacyclic form. doi: 10.1016/j.molbiopara.2018.01.006).

In the Material and Methods section, although the authors referred to a previous published study, important information regarding the parasite strain, and RNA-seq analyses, should be included.

In the conclusion section, I suggest removing the last sentence: “In addition, these findings reinforce previous warnings and mistrust the well accepted nonpathogenic characteristic of the epimastigote stage”.

I suggest several minor changes to be made in the Results/Discussion section:

1- Please add in Figure 1 a quantification of the distinct parasite populations based on the morphology analysis, with the repositioning of the kinetoplast as a criterion to define the transformation into metacyclic forms. This is an important point since, on page 16, the authors indicate that the expression of metacyclic markers “remain constant all along the prolonged starvation of the *T. cruzi* epimastigote culture suggesting that metacyclic trypomastigotes are not significatively contributing to transcriptomic data”. To better discuss this important point, it would be essential do quantify the % of metacyclic forms.

As an alternative, the authors should discuss the possibility that Metacyclin II and III are genes that may be not necessarily good markers for differentiation, since their products have not been characterized.

2- (page 12), please describe more clearly the different combinations of data sets for the comparative analysis of DEGs between the selected time points. Also, it is not clear the meaning of the numbers above the graphs on SFigure2.

3- Data from SFigure 1 did not indicate an increased doubling time for the D28 parasite population, as stated on page 16. Also, SFigure1 should be cited before SFigure 2 in the text.

4- On page 19, the authors wrongly state that among surface proteins, mucins and MASPs constitute the most expressed gene families in *T. cruzi*. However, the largest gene family if Trans-sialidase and TS genes are not only highly expressed in trypomastigotes, but it is also expressed in other parasite stages. This information, added in the following paragraph and should be included in the beginning of this section. Also, the data related to copy numbers of members of TS family encoding active enzymes should be based on other studies (please see Wang et al., 2021, Strain-specific genome evolution in Trypanosoma cruzi, the agent of Chagas disease. doi:10.1371/journal.ppat.1009254; Burle-Caldas et al., 2022, Disruption of active trans-sialidase genes impairs egress from mammalian host cells and generates highly attenuated Trypanosoma cruzi, doi.org/10.1128/mBio.03246-19). Plus, it has been recently shown that, different from what was suggested, TS with enzymatic activity, are not involved in parasite adhesion and invasion of host cells but is mainly involved in parasite egress from the infected cell and survival in the bloodstream of the mammalian host (see Burle-Caldas et al., 2022).

5- On Figure 4, the observation that increased expression, on day 28, of δ-amastins and TcTASV with FC > 10 could be emphasized, not only because it is in accordance with the differentiation to the infective, mammalian stage, but also because it shows that late stationary parasites have the capacity to strongly up-regulate gene expression.

6- Incorrect read mapping may affect the results shown in SFigure 3, since it is well known that short Illumina reads are difficult to map precisely into the different members of TS family, the authors should discuss this possibility. In addition to imprecise mapping, the small values for FC for TS group I and III may indicate that changes in the expression of these genes may not be biologically relevant.

7- Regarding data shown in Figure 6, it is not clear what the author means by this sentence: The presence of the nonreplicative and complement resistant metacyclic trypomastigotes cannot completely explain the huge number of viable cells observed.

8- Data from Figure 7 is impossible to analyze, so please, show larger graphs with numbers of FC that can be read.

## AUTHORS’ RESPONSE TO THE REVIEWERS

Dr. Adeilton Brandão

Handling Editor

Memórias do Instituto Oswaldo Cruz

We are hereby submitting a new revised version of the manuscript. All the reviewer´s concerns were attended and the responses to each of them is presented below. We have agreed with most of the recommendations, acknowledged the suggestions and incorporated them in this new version.

Figure 7 and SFigure 1 were modified as suggested. Besides, Figure 4 and SFigure 3 were improved and the data corresponding to these figures was added also as a new supplementary table (STable II) to highlight the coordinated modulation of multigene families associated with parasite adaptation and persistence within the host. Figure 5 was also redesigned. In addition, a new supplementary figure was added to clarify one concern of both reviewers (SFigure 4).

We would like to thank all for the reviewer process, which has undoubtedly contributed to improving the manuscript and hope you consider this new version suitable for publication

Best regards

Dr. Beatriz Garat

## REVIEWER COMMENTS: Reviewer 1

Reviewer comments: Major comments: The study complements previous findings by the group (Smircich et al., 2023) and intends to present additional evidence supporting the existence of a specific developmental stage of the parasite that, while morphologically resembling an epimastigote, displays distinct biological characteristics—particularly features associated with infectivity. However, the novelty and specific contribution of the present manuscript remain unclear. 

The abstract is generally well written; however, it would benefit from a clearer and more specific articulation of the paper’s main contributions. For instance, in the section corresponding to lines 28 to 40, the description remains vague with respect to the altered gene(s) or gene sets, the nature of these alterations, and how these findings differ from or advance existing knowledge in the literature.

R: The text was modified to include some of the specific genes within the groups: “We found a gene expression early rise of surface protein (such as TS and GP63) and even a rise of TcTASV and δ-amastin, which is not accompanied by increased expression of metacyclic transcript markers. In addition, we found increased expression of genes coding for proteins involved in two other processes activated during the differentiation of epimastigotes to the infective form of the parasite: autophagy (Atg4, Atg7, Atg8.2) and complement resistance (TcCRP and T-DAF).” 

To better contextualize and underscore the relevance of the study’s contributions, I recommend citing and discussing the articles I describe below, which should be incorporated into the manuscript. Although the authors have conducted experimental assays, the notion that epimastigote forms may exhibit differing biological properties has already been proposed by other groups.

R: We certainly agree with the reviewer and consider that we have already clarified this point in the manuscript. Indeed, we conclude that the results we provide here constitute new evidence to support these previous proposals.

For instance, Almeida-de-Faria et al. (1999) (https://doi.org/10.1006/expr.1999.4423), not cited in the manuscript, described an intracellular epimastigote form based on morphometric, biological, and biochemical parameters, as well as ultrastructural antigen localization. This form demonstrates distinct energy and carbon source requirements compared to other intracellular stages of T. cruzi (10.1016/j.molbiopara.2009.07.006, also not cited in the manuscript.

R: Since the work presented in the manuscript is restricted to the insect-dwelling part of the cycle, we have decided not to include the discussion of the papers that distinctly focus on the mammalian part of the life cycle, particularly on an intracellular form. Nonetheless, we have now included a brief comment addressing the reviewer´s concern with the corresponding references. The following text was included: “The plasticity and complexity of T. cruzi forms along the life cycle spread beyond the epimastigote to the metacyclic trypomastigote transition. Transient T. cruzi epimastigote-like forms as intermediates in the differentiation of amastigotes to trypomastigotes inside the mammalian host cells [32] and their distinct energy and carbon source requirements compared to the other intracellular stages [33] have been characterized.” 

Additionally, another study described epimastigote forms derived from metacyclic trypomastigotes—referred to as “recently differentiated epimastigotes”—which possess the capacity to infect mammalian cells and exhibit resistance to complement-mediated lysis (doi:10.1111/mmi.13653, not cited in the manuscript).

R: We thank the reviewer for the recommendation to include this paper. Using cell biology, this paper characterizes the infection ability of recently differentiated epimastigotes obtained from cell derived or metacyclic trypomastigotes. It analyzes, using shotgun, the protein expression of virulence factors of recently differentiated epimastigotes obtained from cell derived trypomastigotes. We have now included this reference in the Introduction section: “In addition, the differentiation from the trypomastigote forms (cell-derived or metacyclic) to an epimastigote-like form named “recently differentiated epimastigotes”, exhibiting complement resistance and infection ability has been recently described using cell biology and proteomic approaches [34].” 

The comparison of the proteomic approach of recently differentiated epimastigotes obtained from cell derived trypomastigotes with the transcriptomic data during prolonged starvation of Trypanosoma cruzi epimastigote related to the acquisition of infective and complement resistance was included in the Results section: “Although, it has been reported that using proteomic approaches for the recently differentiated epimastigotes derived from trypomastigotes [34], the upregulation of some surface proteins such as: a cruzipain protein group (TcCLB.507603.260, TcCLB.507603.270, TcCLB.509429.320 and TcCLB.509429.329), a GP63 (TcCLB.506435.370) and a trans-sialidase (TcCLB.509257.10), no significant differential expression was found for the encoding genes in the transcriptomic data here analyzed.”

A review by De Souza and Barrias (2020) also discusses papers demonstrating that morphologically epimastigote-like forms, termed “transitional epimastigotes”, which are capable of infecting host cells. This same review highlights findings demonstrating that T. cruzi responds to nutritional stress by repressing protein synthesis and protecting mRNA from degradation via sequestration into mRNA granules—evidence of gene expression regulation during early differentiation events. The group itself previously identified differentially expressed genes that serve as markers of the transitional parasite population enriched under nutrient-restricted conditions, supporting the existence of a distinct developmental stage between the recognized insect-dwelling forms. It is important in the presente manuscript to explicitly the genes or gene groups (markers) that meaningfully expand upon previously analyzed data, particularly in the context of nutritional stress response (long starvation). If the aim is to experimentally validate transcriptomic observations and thereby confirm findings derived from in silico analyses, this needs to be clearly distinguished from earlier results. 

R: Both papers were presented in the manuscript. Distinct from De Souza and Barrias, as stated in the manuscript, we use transcriptomic approaches to characterize the differential expression of epimastigotes upon long starvation. Distinctly from our previous paper, as stated in the manuscript, we here focus on genes coding for surface proteins and proteins related to complement resistance and autophagy.

Not only from those mentioned above, but also from classical studies which have already shown that stage-specific gene activation precedes by several days the morphological changes observed during the epimastigote-to-trypomastigote transformation in T. cruzi (e.g., Goldenberg, 1985). Hernandez’s work on stationary-phase T. cruzi epimastigotes has already described morphological and metabolic changes as consequences of nutritional stress. Similarly, studies on metacyclogenesis have established nutritional stress as a key trigger for parasite differentiation. Morphological studies further demonstrate gradual transitions during metacyclogenesis, based on morphological and ultrastructural analyses of differentiating cells. In addition, inhibition of the ubiquitin-proteasome pathway in T. cruzi epimastigotes has been shown to disrupt cell division and differentiation processes, affecting nuclear and kinetoplast DNA distribution. 

R: We have now included the references recommended by the reviewer. These include papers of recognized relevance that constitute basic contributions and/or focus on parasite developmental transitions in different conditions. Globally, all of them together with the results we are presenting certainly contribute to obtaining a more integral landscape of the complex changes the parasite may undergo in the insect vector.

Based on mentioned above, I strongly encourage the authors to revise the manuscript accordingly, or even reconsider to reanalise the data, in light of the studies cited here, which were neither referenced nor discussed. While the manuscript does highlight and discuss gene groups associated with complement resistance, autophagy, and known surface pathogenic factors, these aspects have already been somehow documented in the context of parasite stage differentiation. Maybe the description of genes would be the diferential, but somehow it should be more associated to the biological relevance of the prolonged starvation phase within the T. cruzi life cycle, because it is not clearly addressed. Beyond the microscopy and complement lysis assay, no additional functional experiments were performed. Therefore, it is critical—based on the data presented—to clearly articulate how this work advances the field beyond what has already been published.

R: Following the reviewer’s advice the text was modified to highlight the novelty of the here presented dynamic analysis of transcriptomic changes along prolonged starvation of epimastigotes searching for evidence in our result that could lead to proposing similarities or differences with the published data using different approaches or conditions of metacyclogenesis and/or epimastigogenesis. Following the reviewer’s recommendation, we have also highlighted the behavior of specific genes within the analyzed groups.

Minor comments:

• Line 33, page 10 - “some metacyclic transcript markers”. Should specify which markers.

R: Text was modified to specify the markers: “some metacyclic transcript markers such as metacyclin II and III.”

• Line 6, page 13 - considering discussing in the light of endocytic ability, see the review paper: 10.1016/j.jsb.2016.07.018.

R: Text was modified to include the mentioned reference: “Consequently, repositioning of the kinetoplast and loss of endocytic ability have only been observed in the later stages of the metacyclogenesis process [5, 10.1016/j.jsb.2016.07.018]”

• Line 37, page 19: T.cruzi should be in italic.

R: Text was corrected

• Line 17, page 26 - I suggest to adequate the phrase to “When parasite populations were treated with heat-inactivated complement, a low percentage of PI-positive cells—indicative of cells with disrupted or absent membranes [77]—was detected.”

R: Text was modified accordingly

• Line 38, page 29: “These results complement the distinctive transcriptomic we reported for transitional parasites obtained along the axenic growth of T. cruzi epimastigotes for more than 30 days without nutrient supplementation [25].” I thinks it is still vague, and the authors should more specific in which genes or which data are the complement in comparison to the previus work.

R: Following the reviewer´s observation, the text was modified as follows: “To analyze the transcriptomic dynamics during prolonged starvation of T. cruzi epimastigote culture, we here discriminate early and late response transcripts. In addition, we delved into groups of genes not addressed in our previous work, which are of great importance in the parasite life cycle. We observed a rapid change in surface protein genes such as TS and GP63. Also, the expression of the surface proteins TcTASV and δ-amastin showed upregulation. In addition, an increasing expression of genes coding for autophagy (Atg4, Atg7, Atg8.2) and complement resistance (TcCRP and T-DAF) was found, and the latter was experimentally verified. These results complement the analysis of processes that characterizes the distinctive transcriptomic we reported for transitional parasites obtained along the axenic growth of T. cruzi epimastigotes for more than 30 days without nutrient supplementation…” 

• Sfigure1 and Figure 7 are too small.

R: Thanks for the comment. We have enlarged the indicated figures.

## REVIEWER COMMENTS: Reviewer 2

Reviewer comments: In this work, Perez-Dias et al described a transcriptome analysis based on data that was previously generated by the same group, in which global gene expression of T. cruzi epimastigotes kept in culture for more than 30 days without nutrient supplementation (prolonged starvation) was examined. Distinct from the previously reported analysis, which focused on genes related to protein quality or content, metabolic switch from glucose to amino acid consumption, redox challenge, and surface proteins expression, in this report, the authors focused on other processes related to epimastigotes/metacyclic trypomastigotes differenctiation: parasite autophagy and complement resistance. This is a relevant study that investigates a key process during the T. cruzi life cycle, which contributes to the molecular understanding of parasite differentiation and propagation in the insect vector.

1) The abstract is adequate but could be improved with minor changes that would more faithfully describe the analyses that were performed. As a suggestion, the sentence “Since nutrient restriction is considered one of the main factors driving metacyclogenesis and is very frequent due to the long-term starvation periods that the insect vectors commonly undergo, we have studied the transcriptomic effects of nutrient restriction on long-lasting epimastigote cultures.”, could be changed to “Since nutrient restriction is considered one of the main factors driving metacyclogenesis and is very frequently due to the long-term starvation periods that the insect vectors commonly undergo, we have studied the morphological and growth rate alterations as well as transcriptomic changes that result from nutrient restriction on long-lasting epimastigote cultures.

R: The text was modified following the reviewer´s suggestion.

2) In the Introduction (on page 5), I suggest including a more complete description of the literature regarding RNA-seq analyses published by other groups as well as a few reports describing the role of RNA binding proteins involved with gene expression control during metacyclogenesis.

R: We thank the reviewer for this observation. The citation of these works was incorporated.

3) In the Material and Methods section, although the authors referred to a previous published study, important information regarding the parasite strain, and RNA-seq analyses, should be included.

R: The information on parasite strain T. cruzi Dm28c strain (TcI DTU) was added to the Material and Methods section. We consider that the Materials and Methods section in the manuscript correctly presented the method of analysis of the RNA-seq data from the published paper (open-access) and should not include the method to obtain them (because it is a merit of the published paper).

4) In the conclusion section, I suggest removing the last sentence: “In addition, these findings reinforce previous warnings and mistrust the well accepted nonpathogenic characteristic of the epimastigote stage”.

R: The text was modified following the reviewer´s suggestion.

I suggest several minor changes to be made in the Results/Discussion section:

1- Please add in Figure 1 a quantification of the distinct parasite populations based on the morphology analysis, with the repositioning of the kinetoplast as a criterion to define the transformation into metacyclic forms.

R: Given the characteristics of the transitional forms (displaying a positioning of the kinetoplast, flagellum, and nucleus corresponding to the epimastigote stage but with non-classical epimastigote morphology of the cell body and flagellum) they cannot be precisely quantified by visual inspection. Therefore, as reported in our previous paper (Smircich et al 2023), only total and metacyclic trypomastigote forms were quantified. Following the reviewer´s concern the text was modified to incorporate the quantification of this parasite form: “In these conditions, we have found that the percentage of metacyclic trypomastigote within this long stationary phase displays a composed profile including a gradual increase, not surpassing 10% for a long period (3.8 ± 1.7%, 5.4± 0.4% and 7.9 ± 0.6% for the exponential phase -day 7-, early stationary phase -day 14-, and intermediate stationary phase-day 21 respectively) and then a sharp increase at the end (32.1 ± 5.4% for the final of the stationary phase -day 28-) [35].” 

This is an important point since, on page 16, the authors indicate that the expression of metacyclic markers “remain constant all along the prolonged starvation of the T. cruzi epimastigote culture suggesting that metacyclic trypomastigotes are not significatively contributing to transcriptomic data”. To better discuss this important point, it would be essential do quantify the % of metacyclic forms.

As an alternative, the authors should discuss the possibility that Metacyclin II and III are genes that may be not necessarily good markers for differentiation, since their products have not been characterized.

R: We deeply acknowledge and certainly agree with the reviewer´s observation. The text was modified as follows: “It is worth noting that the profiles of previously reported markers of the metacyclic trypomastigotes, genes coding for Metacyclin II (TcCLB.506529.600) and Metacyclin III (TcCLB.509251.6) [6,27] remain constant all along the prolonged starvation of the T. cruzi epimastigote culture not accompanying the slow increase of metacyclic forms described above (SFigure 2).”

2- (page 12), please describe more clearly the different combinations of data sets for the comparative analysis of DEGs between the selected time points. Also, it is not clear the meaning of the numbers above the graphs on SFigure2.

R: Following the reviewer´s comment we modified the text as follows: “In order to distinguish temporal processes triggered by prolonged starvation within the long stationary phase of T. cruzi epimastigote culture, DEGs were classified as nutrient restriction early response transcripts (ERT) if significant different expression between the data derived from epimastigotes in the exponential (D7) and in the early stationary phase (D14) was observed (subclassified in Figure 2 as following: D14 vs D7, D14 vs D7 plus D21 vs D7, D14 vs D7 and D21 vs D7 and D28 vs D7), likewise, DEGs were classified as late response transcripts (LRT) if the different expression was restricted to the changes between intermediate and late stationary phase (D28 vs D21) (STable I, SFigure 2).”

3- Data from SFigure 1 did not indicate an increased doubling time for the D28 parasite population, as stated on page 16.

R: We agree and thank the reviewer for pointing out this mistake. It should have been decreased instead of increased. The text was modified accordingly.

Also, SFigure1 should be cited before SFigure 2 in the text.

R: SFigure1 is already cited (starting at page 9 l32) before SFigure2 (starting at page 12 l40) in the manuscript. No changes were made.

4- On page 19, the authors wrongly state that among surface proteins, mucins and MASPs constitute the most expressed gene families in T. cruzi. However, the largest gene family if Trans-sialidase and TS genes are not only highly expressed in trypomastigotes, but it is also expressed in other parasite stages. This information, added in the following paragraph and should be included in the beginning of this section.

R: Following the reviewer´s observation the text for mucins and MASPs was changed: “…mucins and MASPs constitute a highly expressed gene family in T. cruzi …” In addition, we added “highly expressed” at the beginning of the TS superfamily paragraph. 

Also, the data related to copy numbers of members of TS family encoding active enzymes should be based on other studies. Plus, it has been recently shown that, different from what was suggested, TS with enzymatic activity, are not involved in parasite adhesion and invasion of host cells but is mainly involved in parasite egress from the infected cell and survival in the bloodstream of the mammalian host.

R: Following the reviewer´s observation, both references were included: “Surface proteins, arranged as large gene families exhibiting diversification and copy number diversity [39], play different roles in the parasite’s life cycle progression, in the host-cell interplay, immune system evasion and persistence of the parasite [40].” “Recently, the TS catalytic activity proposed as a virulence factor has been confirmed and mutants lacking this activity cannot establish infection in mice [52].”

5- On Figure 4, the observation that increased expression, on day 28, of δ-amastins and TcTASV with FC > 10 could be emphasized, not only because it is in accordance with the differentiation to the infective, mammalian stage, but also because it shows that late stationary parasites have the capacity to strongly up-regulate gene expression.

R: We appreciate the suggestion. Following the reviewer´s comment the text was modified: “In summary, the long stationary phase of the epimastigote growth culture during prolonged starvation exhibits some of the molecular characteristics of epimastigotes (TcSMUG and β-amastins) together with some of the metacyclic trypomastigote stage (TS, GP63 and TcTASV) and even δ-amastin being in accordance with the differentiation process but also supporting that late stationary parasites have the capacity to strongly regulate gene expression and suggesting that the intermediate parasite forms may hold cell attachment and invasion potential.”

6- Incorrect read mapping may affect the results shown in SFigure 3, since it is well known that short Illumina reads are difficult to map precisely into the different members of TS family, the authors should discuss this possibility. In addition to imprecise mapping, the small values for FC for TS group I and III may indicate that changes in the expression of these genes may not be biologically relevant.

R: We acknowledge the challenges in accurately mapping reads to highly similar members of large surface protein families. However, these potential biases are consistent across replicates and conditions, ensuring that differential expression analysis remains robust, since any deviation from the true read count would affect both conditions proportionally.

In addition, both Figure 4 and SFigure 3 were redesigned and a new supplementary table (STable II) with data corresponding to Figures 4 and SFigure3 was added to highlight the coordinated modulation of multigene families associated with parasite adaptation and persistence within the host.

7- Regarding data shown in Figure 6, it is not clear what the author means by this sentence: The presence of the nonreplicative and complement resistant metacyclic trypomastigotes cannot completely explain the huge number of viable cells observed.

R: The sentence was removed

8- Data from Figure 7 is impossible to analyze, so please, show larger graphs with numbers of FC that can be read.

R: We appreciate the reviewer’s observation regarding Figure 7. We agree that the graphs were too small to analyze properly. In the revised manuscript, we have increased the size of the panels in the central figure to improve readability. Because our aim with Figure 7 is to illustrate overall trends in gene expression across the pathway, we have also added a supplementary figure (SFigure 4), where the plots are shown at a larger scale. For clarity, each time point’s fold-change (FC) relative to day 7 is indicated on the graph, and statistically significant differences vs. day 7 are denoted with an asterisk.

---

## [Reviewer Report · REVIEWERS COMMENTS]

## REVIEWER #1

I am satisfied with the revised version. The modifications were appropriately incorporated and contributed substantially to improving the manuscript.

## REVIEWER #1

I am satisfied with the revised version. The modifications were appropriately incorporated and contributed substantially to improving the manuscript.